# Bipartite entanglement in a nuclear spin register mediated by a quasi-free electron spin

Marco Klotz [1,3], Andreas Tangemann [1,3], David Opferkuch [1,2] & Alexander Kubanek [1,2] ✉

Quantum networks will rely on photons entangled to robust, local quantum registers for computation and error correction. We demonstrate control of and entanglement in a fully connected three-qubit $^{13}$C nuclear spin register in diamond. The register is coupled to a quasi-free electron spin-1/2 of a silicon-vacancy center (SiV). High strain decouples the SiVs electron spin from spin-orbit interaction reducing the susceptibility to phonons at liquid helium temperature. As a result, the electron spin lifetime of hundreds of milli seconds enables sensing of nuclear-nuclear couplings down to few hertz. To detect and control the register we leverage continuous decoupling using shaped, low-power microwave and direct radio frequency driving. Furthermore, we implement a nuclear spin conditional phase-gate on the electron spin to mediate bipartite entanglement. This approach presents an alternative to dynamically decoupled nuclear spin entanglement, not limited by the electron spin-1/2's nature, opening up new avenues to an optically-accessible, solid-state quantum register.

Solid-state quantum emitters demonstrated first implementations of quantum technologies[1], such as quantum sensors[2,3] and quantum networks[4,5]. Color centers in diamond are of great interest for quantum networking applications due to exceptional optical and spin properties, demonstrating large spin registers with long memory times[6,7], fault-tolerant information processing[8], proof of principal error-corrected spin-photon entanglement[9], distributed networking applications[10] and integration into scalable nanostructures[11].

Group-IV color centers have superior spectral properties as compared to nitrogen-vacancy centers (NV). However, local entanglement of highly coherent $^{13}$C nuclear spins, necessary for error detection and correction, has so far only been demonstrated with spin-1 NV[9]. Challenging operation temperatures of 1.7 K down to few mK to suppress phonon induced dephasing, as well as magnetic field alignment with vector magnets to prevent spin mixing are necessary in the case of group-IV centers[12–15]. Furthermore, the state-of-the-art approach of using dynamical decoupling (DD) of the defect's

electron spin to control nearby nuclear spins [12, 13] is less sensitive for spin-1/2 systems[16]. Very recently, DD was paired with radio frequency control (DDRF), to demonstrate electron-nuclear entanglement and control of two $^{13}$C nuclear spins coupled to a group-IV center[15].

Here, we demonstrate entanglement of two $^{13}$C nuclear spins out of a three-qubit nuclear-spin register coupled to the electron spin of a negatively charged silicon-vacancy center (SiV), schematically depicted in Fig. 1. The ultra-high strain leads to a strong orbital decoupling of the SiV's electron spin, rendering it a quasi-free electron. We operate the system at liquid helium temperature[17] and use efficient low-power microwave and radio frequency driving to directly detect and control the three-qubit nuclear spin register and an additional fourth weakly coupled nuclear spin. We prepare arbitrary register-states with high fidelity by implementing single-shot readout with feedforward[18]. To establish nuclear-nuclear entanglement we utilize the geometric phase acquired during a $2\pi$-rotation of the electron spin conditioned on three nuclear spins[19,20]. Our approach of detecting nuclear spins and

[1]Institute for Quantum Optics, Ulm University, Ulm, Germany. [2]Center for Integrated Quantum Science and Technology (IQST), Ulm University, Ulm, Germany. [3]These authors contributed equally: Marco Klotz, Andreas Tangemann. ✉e-mail: alexander.kubanek@uni-ulm.de

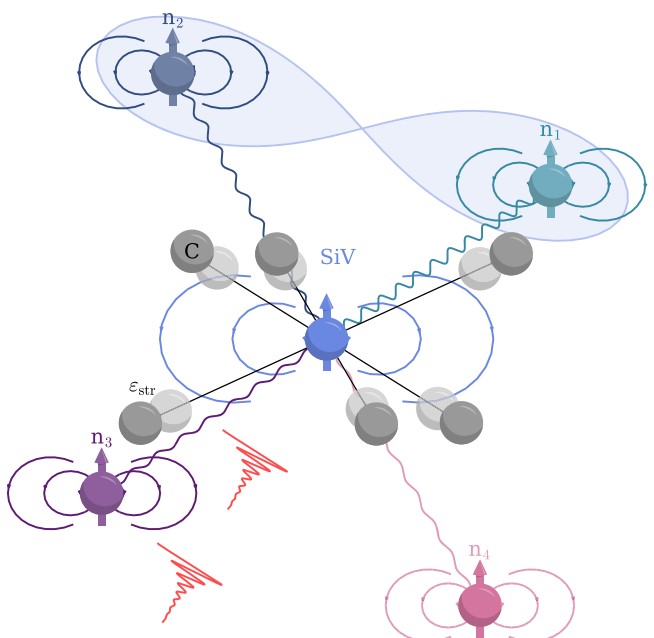

**Fig. 1 | Artistic sketch of a highly strained SiV with a quasi-free electron spin.** The quasi-free electron spin of a highly strained SiV (blue), with opaque and solid circles representing next-nearest carbon (C) atoms, is used as an optically addressable communication qubit. Hyperfine coupled and interconnected $^{13}$C nuclear spins $n_i$ are used as a quantum register with mutual entanglement (shaded blue area).

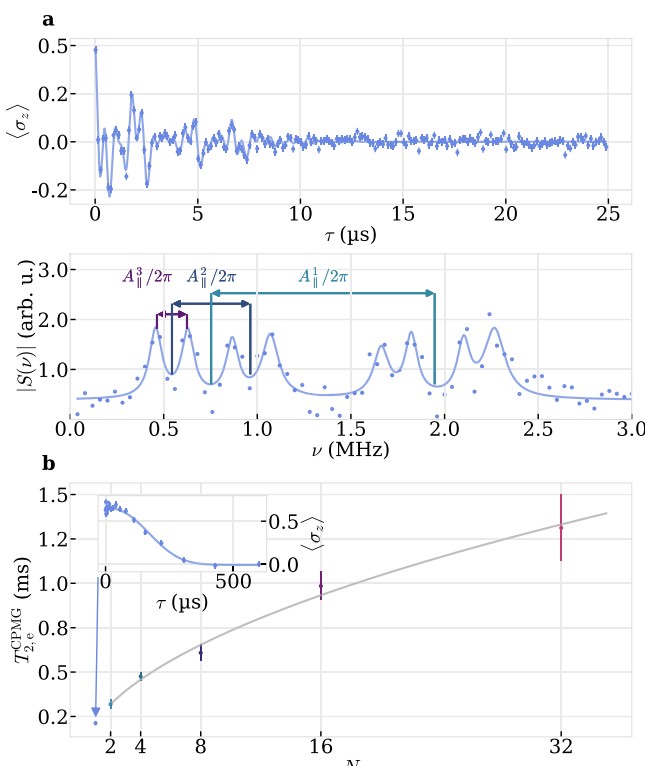

**Fig. 2 | Electron spin coherence and nuclear spin coupling. a** Detuned Ramsey interferometry with inter-pulse spacing $\tau$ on the electron spin (upper panel) with a corresponding Fourier spectrum $S(\nu)$ (lower panel). Solid line in the upper panel is a fit to $\exp(-(\tau/T^*_{2,e})^\beta)\sum^8_{i=1}a_i\sin(\omega_i\tau+\phi^i)+c$. The Fourier spectrum is fitted separately with $S(\nu)=\sum^8_{i=1}a^i\mathcal{L}(\nu,\nu^i_0,\Gamma^i)$, where $\mathcal{L}$, $a^i$, $\nu^i_0$ and $\Gamma^i$ are a Lorentzian, the amplitudes, center frequencies and FWHMs. **b** Scaling $\chi$ of the decoherence time $T^{CPMG}_{2,e}$ as a function of the number of decoupling pulses $N$, i.e. $T^{CPMG}_{2,e}\propto N^\chi$. Inset shows a Hahn echo ($N=1$). We extract the respective decay times from a fit of the form $a\exp(-(\tau/T_{2,e})^\beta)+c$. Error bars represent one standard deviation of the fitting error of each extracted decoherence time.

generating nuclear-nuclear entanglement by means of continuous-decoupling relaxes constraints on the electron spin's hyperfine couplings and coherence time and can instead leverage the long $T^*_{2,n}$ of the nuclear spins.

## Results

### Spin characterization of a quasi-free electron

We are using a SiV hosted in a nanodiamond at liquid-helium temperatures, where strain $\epsilon_{str}$ is largely exceeding the spin-orbit coupling $\lambda_{SO}/2\pi\approx50$ GHz resulting in a ground-state splitting (GS) of $\Delta_{GS}/2\pi=1825.49(39)$ GHz. Causes for high strain include differences in thermal expansion of the sapphire substrate and diamond[14,21] as well as neighboring carbon vacancies[22], where the latter recently reported a SiV with a very similar ground-state splitting. The properties of the sample have been described in detail in ref. 17.

Under such high GS, the SiV's electron spin becomes detached from its orbital degree of freedom, thus approaching a quasi-free electron spin[23,24]. This allows efficient electron spin driving and makes its relaxation time $T_{1,e}$ robust against magnetic field misalignment[24].

Compared to recent results with a SiV with $\Delta_{GS}/2\pi=1111(86)$ GHz[17], we expect that phonon-induced decoherence and relaxation processes between orbital and spin states are further suppressed at our operating temperatures of $T\approx4$ K and static magnetic field of $B_0\approx335$ mT. We polarize the electron spin optically, possible due to an anisotropic g-factor in the optical ground and excited state[25], thereby initializing the electron spin qubit with a fidelity of up to $F_e\approx0.86(1)$, see Methods. Although we measured a relaxation time of $T_{1,e}=0.296(85)$ s and a cyclicity of $\eta=2020(380)$, see Supplementary Fig. 1 [SI], we are unable to profit from single-shot readout of the electron spin to increase the readout fidelity, due to a low photon collection efficiency on the order of 0.1 %.

Compared to the less-strained SiV from[17], this amounts to an increase by three orders of magnitude in $T_{1,e}$, which is expected to also increase nuclear spins' coherence times, since uncontrolled electron spin flips induce decoherence due to different hyperfine precession

frequencies. We characterize the electron spin's coherence properties with coherent microwave (MW) control at a resonance frequency of 9.414 GHz, close to the resonance frequency of a free electron spin 9.388 GHz. The Fourier transform of an off-resonant Ramsey experiment displays eight dominant frequency components, reminiscent of at least three coupled nearby nuclear spins $n_{1-3}$ with parallel hyperfine couplings $A^i_\parallel/2\pi=\{1194(22),418(22),160(22)\}$ kHz and an overall loss of coherence within $T^*_{2,e}=5.69(40)\,\mu$s, see Fig. 2a, setting the resolution of detecting nuclear spins. In order to estimate the probability of finding a resolvable nuclear-spin configuration with variable size, we used Monte-Carlo simulations in Suppl. Note 2.

We additionally perform a coherent population trapping (CPT) experiment and independently verify $A^1_\parallel/2\pi=1144(31)$ kHz and $A^2_\parallel/2\pi=415(27)$ kHz with a resolution of $\Gamma^{CPT}=149(24)$ kHz, see Supplementary Fig. 3 [SI]. Using a single refocusing MW-pulse, i.e. a Hahn echo, the spin's coherence time can be extended to $T^{Hahn}_{2,e}=213(12)\,\mu$s, see inset of Fig. 2b. Increasing the number of decoupling pulses to $N=32$ in a CPMG type of measurement, the coherence time can be further extended to $T^{CPMG}_{2,e}=1.31(19)$ ms.

The resulting scaling, $T^{CPMG}_{2,e}\propto N^\chi$, is well described with a scaling factor $\chi=0.513(26)$, shown in Fig. 2b.

We attribute the dominant noise source to a bath of free electrons potentially originating from surface defects of the host or other defect centers in the vicinity[23]. To investigate the spin-bath further, we performed double electron-electron resonance (DEER), see Supplementary Fig. 4b [SI].

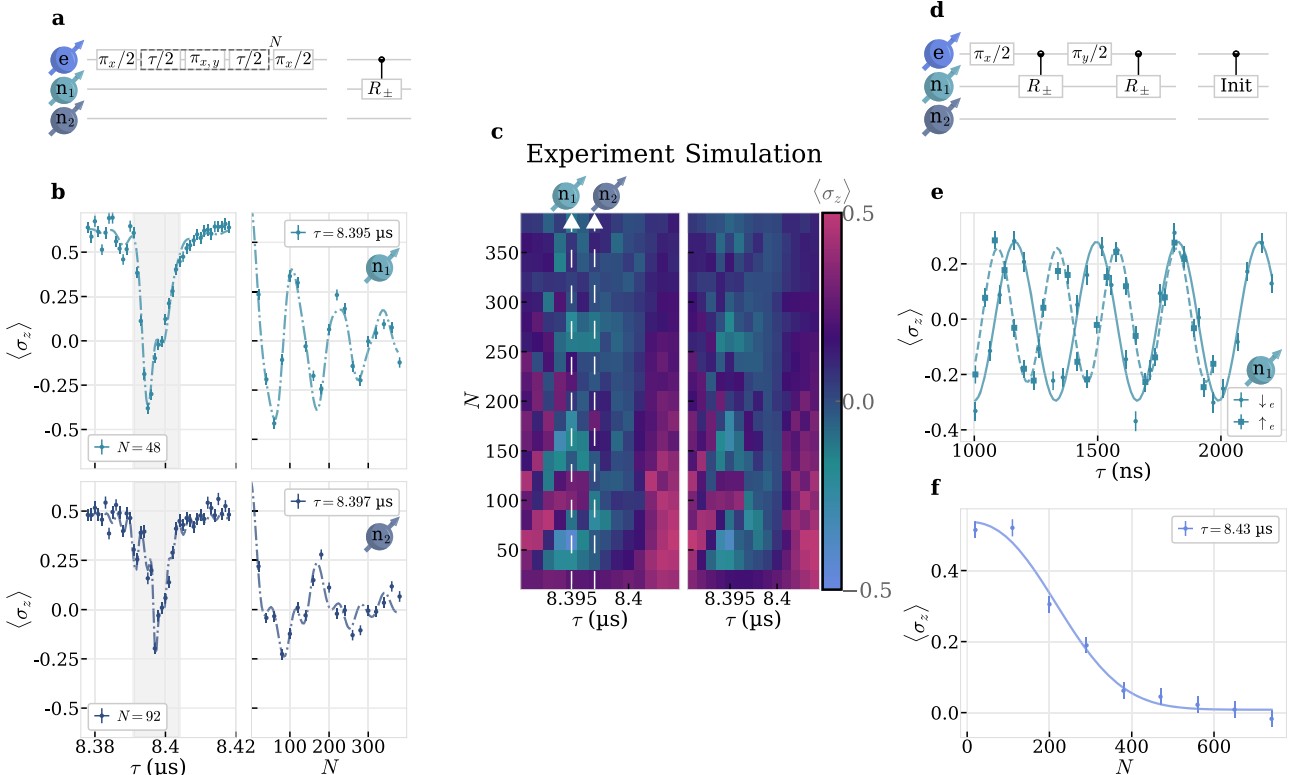

**Fig. 3 | Dynamically decoupled nuclear spin detection and control. a** Pulse sequence for dynamically decoupled nuclear spin rotation conditional on the electron spin $R_\pm(N, \tau)$[26]. $\pi_{x,y}$-elements are rotations around orthogonal axes, $\tau$-elements indicate free precession intervals and $N$ is the number of repetitions. **b** An inter-pulse spacing $\tau$ sweep at $N = (48, 92)$, upper and lower left panel, shows dominant resonances of two nuclear spins, $n_1$ and $n_2$, where the electron spin's coherence is inverted at $\tau = (8.395\,\mu s, 8.397\,\mu s)$. A subsequent sweep of $N$ at the resonant $\tau$ shows coherent oscillations, upper and lower right panel. Dash-dotted lines are a simulations using the numerical model fitted with the 2D dataset. **c** A 2D ($\tau, N$) sweep around the two resonances from (**b**), indicated with white lines, shows

the transition of strong coherent coupling to the weakly coupled bath. Left is the measured data from which we extract system parameters with a numerical model. The fitted model is shown on the right. **d** Nuclear spin initialization sequence adapted from refs. 23,28. **e** Electron spin-dependent nuclear Ramsey inter-ferometry by initialization of one nuclear spin with the sequence in (**d**) and per-forming a $\pi/2$-rotation with the sequence in (**a**). For the targeted nuclear spin $A^1_\perp/2\pi = 233.19(81)$ kHz we measure $A^1_\parallel/2\pi = 1111(48)$ kHz. **f** Off-resonant DD show-ing a loss of coherence after $N = 274(17)$ ($T^{XY}_{2,e} = 2.31(14)$ ms), fitted with a stretched exponential decay, solid line.

We find a spectrally broad resonance at 9.3889(38) GHz, which agrees well with the Larmor frequency of a free electron spin. Additionally, we find a more prominent and sharp resonance in the DEER spectrum at 9.43064(77) GHz, which requires further investigation.

**Dynamically decoupled nuclear spin detection and control**
In order to leverage highly coherent $^{13}$C nuclear spins from the sur-rounding lattice as a quantum register, individual detection and coherent control is required. One well-established method to char-acterize the nuclear spin environment relies on dynamically decou-pling (DD) the electron spin from the surrounding nuclear spin bath except for specific target nuclear spins. Under the right conditions, i.e. the right inter-pulse spacing $\tau$ and number of $\pi$-pulses $N$ in the pulse sequence depicted in Fig. 3a, entanglement of that nuclear spin with the electron spin leads to a loss of electron spin coherence[23,26]. Due to the spin-1/2 nature of the SiV's electron spin, this approach is only sensitive to second order in perpendicular hyperfine coupling, $A_\perp$, and nuclear Larmor frequency, $\omega_{L,n}$, i.e. $(A_\perp/\omega_{L,n})^2$[16,27], therefore necessi-tating large $\tau$ to separate resonances of different nuclear spins, requiring long coherence time.

In Fig. 3b we sweep $\tau$ (left) and $N$ (right) of an XY-($\tau, N$) DD sequence around an exemplary resonance indicating entanglement with various nuclear spins. Clearly visible are the oscillations of the electron spin's coherence with $N$ at different $\tau = 8.395\,\mu s$ and 8.397 s for the most prominent resonances.

We extend the measurement to a full 2D sweep of $\tau$ and $N$ around the resonance and use a numerical model including the central electron spin, three target and one parasitic nuclear spin to fit the 2D dataset with good agreement to the experimental data, see Fig. 3c. This allows us to extract the bare Larmor frequency $\omega_{L,n}/2\pi = 3.582518(11)$ MHz as well as the perpendicular hyperfine coupling components of the strongest and parasitic spins $A^i_\perp/2\pi = \{233.19(81), 147.7(15), 75.5(35), 46.8(44)\}$ kHz. See Methods for further explanation of the used model and parameters.

We verify exemplarily that the nuclear spin with $A^1_\perp$ belongs to $A^1_\parallel$ by using a tailored DD initialization sequence from[17,23,28], requiring electron spin conditioned $\pi/2$-rotations of the target nuclear spin with a slightly off-resonant $\tau_{init}$, depicted in Fig. 3d. We implement the corresponding sequence with $N = 24$ and sweep $\tau_{init}$ around the reso-nance, where individual initialization is achieved at $\tau_{init,1} = 8.3915\,\mu s$, Supplementary Fig.5 [SI]. Using the initialization sequence together with a conditional $\pi/2$-rotation, implemented with the pair ($\tau_{CX} = 8.395\,\mu s$, $N_{CX} = 24$), we probe electron-spin dependent nuclear Ramsey interferences, depicted in Fig. 3e, which reveal two distinct precession frequencies split by 1111(48) kHz, in good agreement with $A^1_\parallel$ extracted from electron Ramsey measurements.

The procedure could be extended to the other two nuclear spins in a future study. In order to probe the electron spin's coherence under XY-$N$ DD we choose $\tau = 8.43\,\mu s$, off-resonant with a multiple of the nuclear spins precession period, sweep $N$ and observe an exponential

decay in coherence within $T_{2,e}^{XY} = 2.31(14)$ ms, shown in Fig. 3f. The measured $T_{2,e}^{XY}$ poses an ultimate limit to our sensitivity of detecting nuclear spins with DD and limits indirect nuclear spin control gates, as can be seen by the exponentially decaying oscillations in Fig. 3b, right panel. It is also worth noting that $T_{2,e}^{XY}$ differs from an extrapolation of the CPMG measurements, which yields $T_{2,e}^{CPMG,274} = T_{2,e}^{CPMG,32} \cdot (274/32)^\chi \approx 3.9$ ms. The discrepancy is attributed to the specific noise characteristics at $\tau = 8.43\,\mu$s or pulse errors at large $N \gg 32$. Addressing more nuclear spins using DD sequences on the electron spin would require either a more favorable ratio of $A_\perp^i/\omega_{L,n}$ or longer inter-pulse spacings to separate more weakly coupled nuclear spins from the bath. However, because of the high strain we rely on a strong magnetic field and therefore large $\omega_{L,n}$ to still retain a high electron spin initialization fidelity. For this reason, we switch to a more direct approach to nuclear spin detection and control.

### Direct nuclear spin spectroscopy

The strain-induced strong decoupling of orbital and spin degree of freedom protects the electron spin's transition frequency from shifts due to strain fluctuations which can be present in nanohosts[23,24]. In addition, the use of cryogenically cooled permanent magnets further reduces external magnetic field fluctuations, hence allowing us to drive Rabi oscillations with low-power MW-pulses while continuously decoupling. For example, Fig. 4a shows multiple Rabi oscillations with $\Omega_{R,e}/2\pi = 70.93(59)$ kHz and $T_{2,e}^{Rabi} = 51(10)\,\mu$s, an order of magnitude longer than $T_{2,e}^*$, enabled by continous DD of the electron spin[29]. We can measure Rabi oscillations down to 5.49(22) kHz with $T_{2,e}^{Rabi} = 161(30)\,\mu$s, see Supplementary Fig 6 [SI].

We scan for various nuclear spin-dependent resonances sweeping the frequency $\nu_{MW}$ of a $\pi$-pulse with the previously determined Rabi frequency of $\Omega_{R,e}/2\pi = 70.93(59)$ kHz, setting the lower bound for the spectral resolution. Fig. 4b shows a spectrum of eight distinct resonances of the previously found nuclear spins, also visible in Fig. 2a, from which we extract $A_\parallel^i/2\pi = \{1181.9(71), 411.5(71), 144.6(71)\}$ kHz with a full-width-half-maximum (FWHM) of $\Gamma = 134.8(22)$ kHz $> \Omega_{R,e}/2\pi$, hinting towards the presence of a fourth nuclear spin which splits the eight resonances further but is undetectable.

Interestingly, if we polarize the three strongest nuclear spins, see next section, and perform the same measurement with a lower Rabi

frequency of $\Omega_{R,e}/2\pi = 7.7$ kHz the resolution increases and we can tentatively assign a fourth splitting with $A_\parallel^4/2\pi = 32.5(15)$ kHz with a width of 43.0(23) kHz which is below the inhomogeneous linewidth $\Gamma_{2,e}^*/2\pi = C(\beta) \cdot 1/(\pi T_{2,e}^*) = C(\beta) \cdot 56.0(39)$ kHz, where $C(\beta) = 1(2\sqrt{\ln 2})$ for a stretch factor $\beta = 1(2)$ in the Ramsey measurement, see Fig. 4c.

A reduction in dephasing rate upon polarization of the nuclear spin bath has also been observed for NV- centers in diamond[30] and quantum dots[31]. Moreover, we verify the presence of a fourth nuclear spin by resolving a beat tone in a low-power Rabi measurement with resonant and effective Rabi frequency $\Omega_{R,e}/2\pi = 19.50(25)$ kHz and $\Omega_{R,e}^{eff}/2\pi = 43.0(14)$ kHz, reminiscent of a detuning of $\Delta/2\pi = 38.65(85)$ kHz, close to the $A_\parallel^4/2\pi$, see Supplementary Fig. 6 [SI].

### Radio frequency nuclear spin control

Using the extracted $\omega_{L,n}$ and $\{A_\parallel^i, A_\perp^i\}$, we can search more directly for the coupled spins' resonances and coherently control them. We realize a nuclear-spin controlled electron flip $C_{\Uparrow_i/\Downarrow_i}NOT_e$ for all four nuclear spins by consecutively driving the respective combinations of electron spin resonances with sinc-shaped MW-pulses to increase spectral selectivity. We truncate the sinc-pulses at the second zero crossing to reduce the temporal extend of the pulses. The bandwidths of the $C_{\Uparrow_i/\Downarrow_i}NOT_e$ are chosen to be close to each $A_\parallel^i/2\pi$, respectively, see Fig. 5a, b and Suppl. Note 7 [SI]. To realize electron-spin controlled nuclear flips $C_{\uparrow/\downarrow}NOT_{n_i}$ we use rectangular radio frequency (RF) pulses.

We initialize and read out each nuclear spin $n_i$ by swapping the electron and nuclear spins populations with a $C_{\Uparrow_i}NOT_e$ and a $C_\uparrow NOT_{n_i}$ before reading out the electron spin, see Fig. 5c. By inverting the electron spin with an optional $\pi$-pulse and changing the respective RF frequency, we can drive coherent nuclear Rabi oscillations with Rabi frequencies $\Omega_{R,n_i}/2\pi$ from 3.564(13) kHz to 5.069(17) kHz on all four nuclear spins conditional on the electron spin $|\uparrow\rangle/|\downarrow\rangle$, circuit diagram of the sequence and measurement data are shown in Fig. 5c, d. The offset and decreasing contrast with decreasing $A_\parallel$ is attributed to limited $C_{\Uparrow_i/\Downarrow_i}NOT_e$ fidelities ranging from 0.96 for $n_2$ to 0.53 for $n_4$, most likely arising from either pulse errors due to a detuning or onset of decoherence since the pulses get longer with decreasing bandwidth. See Supplementary Fig. 7 [SI] for measurement data of the sinc-pulses.

These interpretations can be further verified by considering simulations done with the previously established numerical model consisting of one electron and four nuclear spins, see dash-dotted lines in Fig. 5d.

It is worth noting that we observe a difference in Rabi frequencies which is higher for larger hyperfine couplings, possibly due to an electron spin-induced modification of the nuclear spin's magnetic moment[32], and not reflected in the numerical model, where we explicitly included the measured Rabi frequencies. Additionally, we deliberately used a lower $\Omega_{R,n}$ for $n_4$, since we observe a beat tone when driving with $\Omega_{R,n_4}/2\pi \approx 4$ kHz potentially from unwanted polarization and driving of a fifth nuclear spin, see Supplementary Fig. 8 [SI].

Ramsey measurements with detuning $\Delta/2\pi \approx 500$ Hz on each $n_i$ yield typical dephasing times of $T_{2,n_i}^* = \{3.72(25), 5.39(31), 4.87(54), 8.1(89)\}$ ms[7,15], see Fig. 5 e, f.

We further investigate the connectivity of our nuclear spin register using spin-echo double resonance (SEDOR) measurements on the three strongest coupled nuclear spins[7]. We perform a Hahn echo on the sensor nuclear spin ($s$) to increase the coherence time and therefore sensitivity. Simultaneous to the refocusing $\pi$-pulse on the sensor, we apply a $\pi$-pulse on the target nuclear spin ($t$). As a result, we can accumulate the phase from a mutual coupling $C_{s,t}$ between the two spins, whereas every other nuclear spins' couplings are rephased. Fig. 5g shows exemplarily the sequence for $n_s = n_2$ and $n_t = n_1$. The measured oscillations in the nuclear spin coherence, see Fig. 5h are a result of mutual coupling $C_{s,t}$ between the spins. Due to the electron

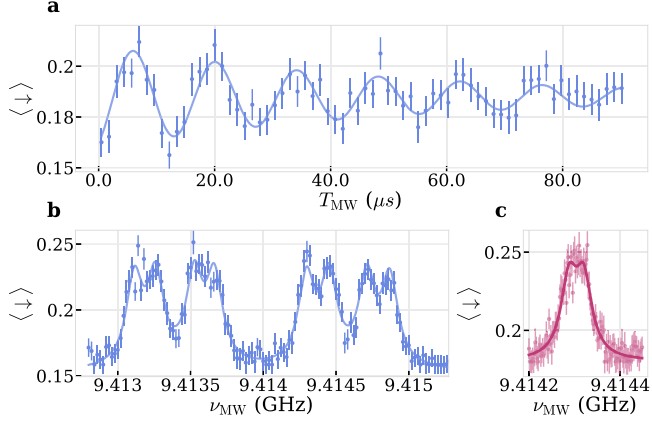

**Fig. 4 | Low-power microwave sensing. a** Continuously decoupled low-power Rabi driving showing persistent oscillations at a frequency of $\Omega_{R,e}/2\pi = 70.93(59)$ kHz beyond the dephasing time $T_{2,e}^* = 5.69(40)\,\mu$s. Solid line is a fit to $a\sin(\Omega_{R,e}T_{MW} + \phi)\exp(-T_{MW}/T_{2,e}^{Rabi}) + c$ **b** Frequency scan of a $\pi$-pulse with $\Omega_{R,e}$ from (**a**) reveals eight resonances which are fit with $a\sum_{i=1}^8 \mathcal{L}(\nu_{MW}, \nu_0^i, \Gamma) + c$ (solid line), where $\mathcal{L}(\nu_{MW}, \nu_0^i, \Gamma)$ is a Lorentzian. **c** Polarizing the three nuclear spins from (**b**) before scanning the frequency at $\Omega_{R,e}/2\pi = 7.7$ kHz increases sensitivity below the unpolarized inhomogeneous linewidth and shows a fourth nuclear spin with $A_\parallel^4/2\pi = 32.5(15)$ kHz, extracted from a double-Lorentzian (solid line).

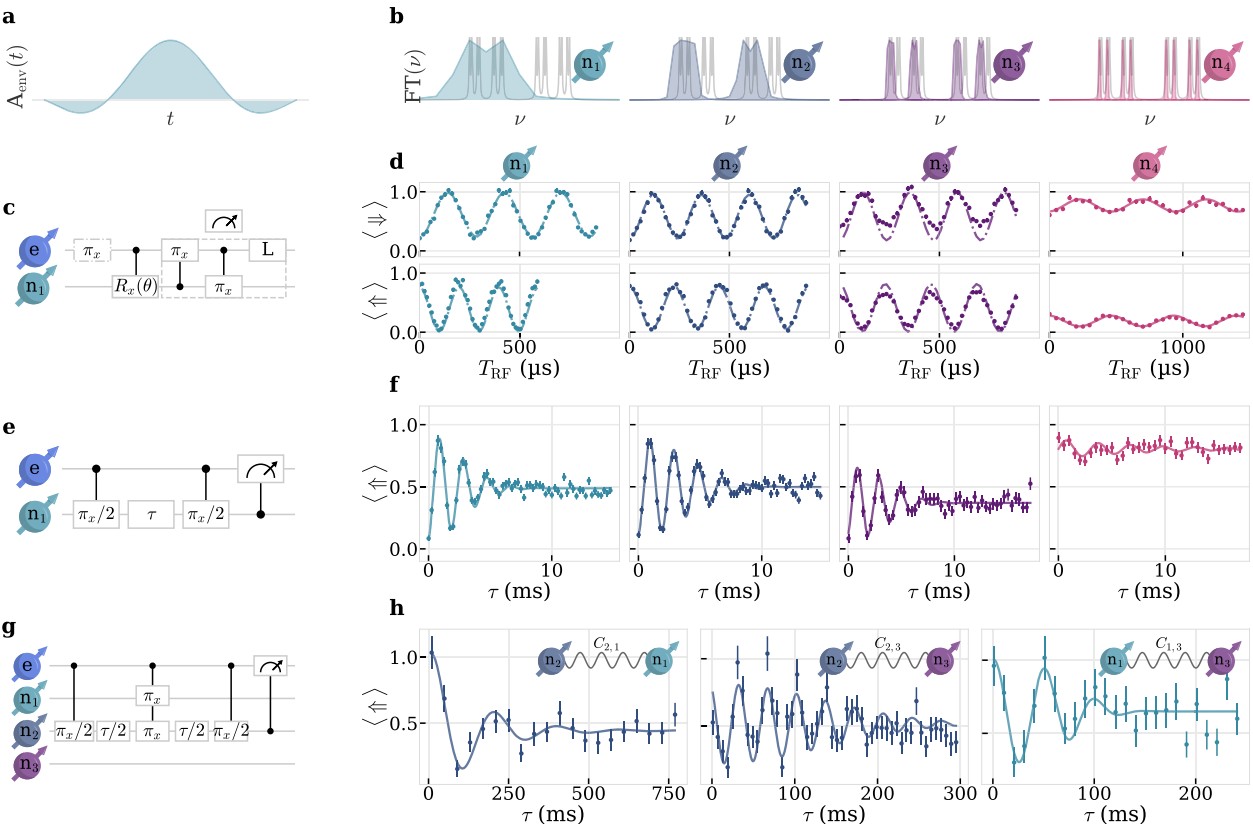

**Fig. 5 | Radio frequency control and characterization of four nuclear spins.**
**a** Temporal and **b** spectral shape of the truncated sinc-pulses to realize the four
different $C_{n_i}NOT_e$, where $n_i = |\Downarrow_i\rangle$. **c** Pulse sequence to rotate and swap a desired
nuclear spin with the electron spin followed by a readout laser-pulse. **d** Electron
spin-dependent Rabi oscillations of the populations $|\Uparrow\rangle/|\Downarrow\rangle$, read out with the
respective $C_{\Uparrow_i/\Downarrow_i}NOT_e$. Upper/lower row indicate measurements where the elec-
tron is in $|\uparrow\rangle/|\downarrow\rangle$, necessary to realize both $C_{\uparrow/\downarrow}NOT_{n_i}$ on all four nuclear spins.
Dash-dotted lines are obtained from simulations of our numerical model, see

"Methods". Solid line is a fit to $a_i \sin(\Omega_{R,n} T_{RF} + \phi) \exp(-(T_{RF}/T_{2,n}^{Rabi})) + c$ **e** Pulse
sequence and **f** experimental data for nuclear Ramsey interferometry on all four
nuclear spins with fits to $a \sin(\Delta\tau + \phi) \exp(-(\tau/T_{2,n}^*)^\beta) + c$ (solid lines). **g** Pulse
sequence for SEDOR experiment involving $n_2$ as sensor and $n_1$ as target spin.
**h** Measurement of the mutual coupling $C_{s,t}$ between the three strongest-coupled
nuclear spins, with the respective sensor (*s*) and target (*t*) spin. Solid lines are fits
to $a \sin(C_{s,t}\tau + \phi) \exp(-(\tau/T_{2,s}^{SEDOR})^\beta) + c$.

## Table 1 | Register connectivity measured with SEDOR. s,t : sensor/target nuclear spin

| s,t | $C_{s,t}/2\pi$ | $T_{2,n}^{SEDOR}$ |
|-----|----------------|-------------------|
| 2,1 | 5.12(36) Hz | 0.137(32) s |
| 2,3 | 28.56(35) Hz | 0.181(32) s |
| 1,3 | 19.7(12) Hz | 0.090(19) s |

$C_{s,t}$: coupling strength between s and t. $T_{2,n}^{SEDOR}$: Coherence time extracted from SEDOR
measurement.

spin's $T_{1,e}$ the phase accumulation of the sensor nuclear spin gets
randomized, limiting sensitivity to few Hz. The coupling components
and coherence times are listed in Table 1. Having determined all RF
resonance frequencies $\omega_{\uparrow/\downarrow,n_i}/2\pi = \sqrt{(\omega_{L,n} \pm A_\parallel^i/2)^2 + (\pm A_\perp^i/2)^2}/2\pi$,
we can independently extract the hyperfine parameters of the three
strongest nuclear spins $A_\parallel^i/2\pi = \{1194, 420, 141\}$ kHz and
$A_\perp^i/2\pi = \{242, 174, 98\}$ kHz which is more accurate compared to the fit
of the 2D XY-*N* measurements, since we are here only limited by the
$T_{2,n}^*$ of the nuclear spins. Using the same method we extract
$A_\parallel^4/2\pi = 34$ kHz, inline with previous low-power Rabi measurements on
the electron spin.

In order to improve and overcome the electron-spin limited
initialization fidelities of each $n_i$ realized by the previous swap
sequence, we implement initialization by measurement. We readout

each nuclear spin's state by applying the corresponding aforemen-
tioned $C_{\Uparrow_i/\Downarrow_i}NOT_e$ together with a 60 $\mu$s readout laser-pulse and repeat
it for a total laser-on time of $T_{SSR} = 5$ ms. To prevent nuclear spin
polarization[17,18] we apply the sequence alternately on both nuclear
spin states by switching the $C_{\Uparrow_i/\Downarrow_i}NOT_e$ frequencies from $|\Downarrow_i\rangle$ to $|\Uparrow_i\rangle$.
We then count the number of collected photons within the single-shot
readout (SSR) window and make a histogram of the photon counts, see
first row of Fig. 6a for nuclear spins $n_1 - n_3$. In this way we can dis-
criminate dark and bright states by choosing a threshold, indicated in
Fig. 6a by a dashed line, which minimizes the mutual overlap of two
Gaussian distributions. This leads to fidelities $F_{n_i} \approx \{0.98, 0.98, 0.96\}$
for $n_{1-3}$[17].

Moreover, we can deterministically prepare arbitrary nuclear
register states with active-feedback by controlling the application of an
RF $\pi$-pulse on $n_i$ conditioned on the number of detected photons in a
SSR window. We count the photons within a SSR windows with a FPGA
counter which conditionally on the counted photon number blocks
the application of a subsequent RF $\pi$-pulse. In the second and third row
of Fig. 6a the measured photon statistics for the two different
$C_{\Uparrow_i/\Downarrow_i}NOT_e$ readout frequencies are shown for each nuclear spin,
where active feedback prepares $n_1$-$n_3$ in a specific state. As an example,
we prepare the register state $|\Uparrow_1\Uparrow_2\Uparrow_3\rangle$ and apply multi-conditioned
150 kHz $\pi$-pulses on the eight electron spin transitions, thereby read-
ing out the register's state. For the $|\Uparrow_1\Uparrow_2\Uparrow_3\rangle$ resonance we measure a
population of 0.812(25), see Fig. 6b, which corresponds to an average
nuclear spin initialization of 0.929(18) of each nuclear spin, corrected

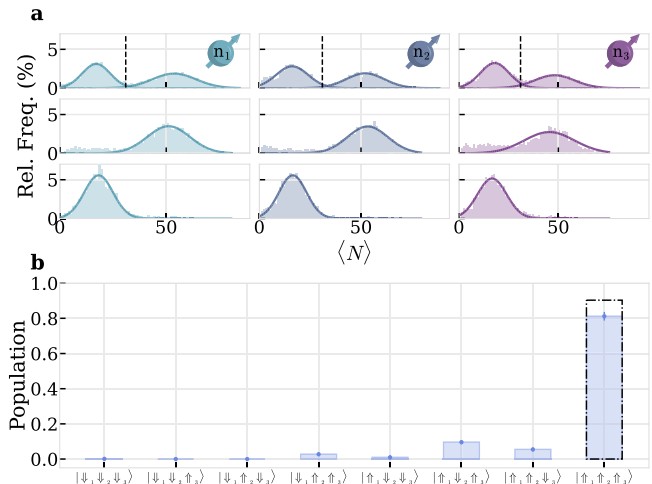

**Fig. 6 | Single shot nuclear spin readout. a** Photon counting statistics of repetitively applying a $C_{\Uparrow_i/\Downarrow_i}NOT_e$ with a short laser for a total laser on time $T_{SSR} = 5$ ms. Top row shows bare histogram without feedback. Center and bottom row show distributions with feedback and active control of an RF $\pi$-pulse after the SSR window to prepare a desired nuclear spin state, where each column indicates $n_1$ to $n_3$ from left to right. **b** Deterministic preparation of register state, $|\Uparrow_1\Uparrow_2\Uparrow_3\rangle$ in this example, by applying a SSR to each $n_i$ with active feedback. Readout on each resonance is done with a 150 kHz sinc-pulse. Dash dotted bar indicates the expected contrast with nuclear spin initializations as determined in (**a**).

for the readout infidelity of the 150 kHz pulse. We attribute the discrepancy to the measured individual fidelities $F_{n_i}$ to instabilities of the experimental setup which lead to drifts in photon-collection efficiency and therefore misclassifications after the SSR windows.

## Phase-gate and nuclear-nuclear spin entanglement

In order to use the three-qubit register as a quantum memory with error detection, we need at least two entangled qubits. Additional single quantum-error correction would require three-partite entanglement[9,19,26]. In general, entanglement between two nuclear spins, i.e., $n_1$ and $n_2$, can be established by using dynamical decoupling to first create a Bell-state between the electron spin and $n_1$ with an additional swap of the electron onto $n_2$[9,26]. This approach is limited by the decoherence of the electron spin, since it is in a superposition during register state preparation, as well as infidelities originating from coupling to other nuclear spins due to the electron spin-1/2's unfavorable resonance condition proportional to $(A_\perp^i/\omega_{L,n})^2$.

Here, in order to circumvent limited coherence of the electron spin, we take a different approach entangling two nuclear spins[19]. We take advantage of the geometric phase $\gamma_g = \Omega/2$ acquired during a full rotation of the electron spin which is given by half the solid angle $\Omega = 2\pi\left(1 - \Delta/\sqrt{\Delta^2 + \Omega_{R,e}^2}\right)$ enclosed by the trajectory on the Bloch sphere, where $\Delta$ is the detuning[33]. In the resonant case, i.e. $\gamma_g \overset{\Delta=0}{=} \pi$, we can thus construct a $\pi$-phase-gate conditioned on the nuclear spins' state (CPhase) by driving the respective resonance with a $2\pi$-rotation, see Fig. 7a. The circuit diagram of the entangling sequence is depicted in 7b. Starting with a SSR on all $n_i$ to prepare the register in $|\Uparrow_1\Uparrow_2\Uparrow_3\rangle$, we then post-select on $|\Uparrow_1\Uparrow_2\Uparrow_3\rangle$ with an additional SSR window only

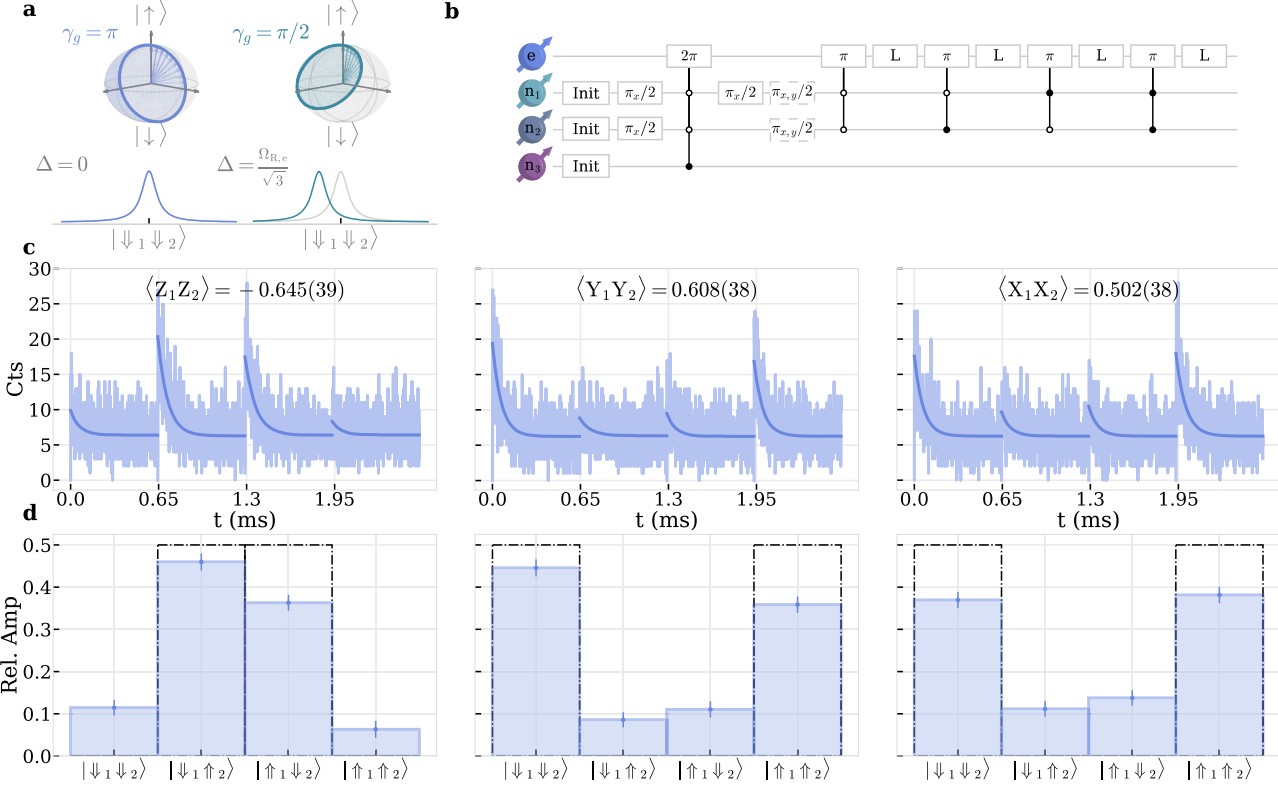

**Fig. 7 | Nuclear-nuclear entanglement generation. a** Geometric phase $\gamma_g = \Omega/2$ acquired during a full $2\pi$ evolution of the electron spin which encloses solid angle $\Omega$ and is conditioned on $|\Downarrow_1\Downarrow_2\rangle$. Left(right) panel shows a resonant(detuned) case with solid angle $\Omega = 2\pi(\pi)$, respectively. **b** Circuit diagram for generating a Bell-state between $n_1$ and $n_2$. Init blocks consist of three SSR windows followed by an additional post-selection, see text. $L$ is a read-out laser-pulse. Full(hollow) circles are conditional gates on $|\Uparrow\rangle(|\Downarrow\rangle)$, respectively. Dashed line indicate additional single-qubit gates to rotate the readout basis. **c** Photon histogram during read-out lasers of $|\Downarrow_1\Downarrow_2\rangle$, $|\Downarrow_1\Uparrow_2\rangle$, $|\Uparrow_1\Downarrow_2\rangle$ and $|\Uparrow_1\Uparrow_2\rangle$ resonances. The different columns show the signal after application of additional basis rotations, see (**b**). **d** Relative readout amplitudes of (**c**) $a_i/\sum_i a_i$, reflecting the populations of respective nuclear spin states. Black dash-dotted line show the populations and coherences of an ideal Bell-state $(|\Downarrow_1\Uparrow_2\rangle + |\Uparrow_1\Downarrow_2\rangle)/\sqrt{2}$.

on that resonance to further improve register initialization to 0.94, see Supplementary Fig. 9 [SI]. After that, we apply unconditional single-qubit $\pi/2$ rotations on $n_1$ and $n_2$ using simultaneous driving of $\omega_{\Uparrow/\Downarrow, n_i}/2\pi$, followed by a CPhase-gate conditioned on $|\Downarrow_1\Downarrow_2\Uparrow_3\rangle$ with a 150 kHz sinc-pulse. The last $\pi/2$-pulse on $n_1$ results in the Bell-state $(|\Downarrow_1\Uparrow_2\rangle + |\Uparrow_1\Downarrow_2\rangle)/\sqrt{2}$ between $n_1$ and $n_2$. It is worth noting, that the electron spin has not changed state. We finally read out $n_1$ and $n_2$ by applying 500 kHz sinc-pulses at the respective resonance frequencies followed by a laser-pulse, see Fig. 7b. Additional single-qubit gates, phase-shifted by 0° and 90° compared to the initial $\pi/2$-phase, rotate the readout basis to extract the coherences. From the respective normalized amplitudes $a_i/\sum_i a_i$ we infer the relevant correlators $\langle Z_1 Z_2 \rangle$, $\langle Y_1 Y_2 \rangle$ and $\langle X_1 X_2 \rangle$, see Fig. 7c and d. We then extract a state fidelity to the corresponding Bell-state by calculating $F = (\langle I, I \rangle + \sum_{i=ZZ,YY,XX} c_i \langle i \rangle)/4$, where $c_i$ depend on the prepared Bell-state.

In the given case of $(|\Downarrow_1\Uparrow_2\rangle + |\Uparrow_1\Downarrow_2\rangle)/\sqrt{2}$ the correlator coefficients are $c_{ZZ} = -1$, $c_{YY} = +1$ and $c_{XX} = +1$ which results in $F = 0.689(17)$, demonstrating preparation of an entangled two-qubit state. See Suppl. Note 10 [SI] for the other Bell-States. Potential error sources to the fidelity are almost entirely dominated by the erroneous $2\pi$ gate fidelity of the 150 kHz sinc-pulse which we set equal to the previously mentioned $\pi$-pulse fidelity 0.73, see Supplementary Fig.7 [SI]. Another relatively minor infidelity arises due to the limited readout fidelity 0.96 of the 500 kHz-pulse. Combined, these two effects set an upper bound for the reachable Bell-state fidelity of $F \approx 0.7$.

## Discussion

In conclusion, a highly strained SiV and the resulting strong orbital decoupling yields a quasi-free electron spin with preserved coherence over milliseconds at elevated temperatures of 4K. Operation with highly stable permanent magnets and low-power MW driving allows us to control a three-qubit nuclear-spin register and a fourth weakly coupled nuclear spin. We demonstrate bipartite entanglement of highly coherent $^{13}$C nuclear spins. Entanglement mediation is achieved by leveraging a nuclear-spins' conditional geometric phase gate on the electron spin. We realize this with continuously decoupled, spectrally selective microwave pulses which also enable high-fidelity single-shot readout of the three-qubit nuclear spin register.

Our findings show that control and entanglement of a multiqubit $^{13}$C spin register is not limited by the spin-1/2 and hence can be directly translated to other optically-active, spin-1/2, solid-state defects[4,12,14,34–37].

In future experiments, the entanglement can be further improved by using optimal control to increase the fidelity of the CPhase-gate, as has been shown with NV centers[19,20]. This will enable entanglement of all three nuclear spins to implement a logical memory qubit with error correction capabilities. Furthermore, the presented SEDOR sensitivity down to few Hz will allow us to work with weakly coupled nuclear spins to increase the register size toward a fully fault-tolerant five-qubit memory[2]. The so far limited electron spin initialization fidelity can be improved by increasing the magnetic field strength without drawbacks for the presented direct nuclear spin control and entangling scheme. Integration of the color center's nanohost into photonic structures will improve collection efficiency and allow electron-spin single-shot initialization by measurement[38,39].

We believe that combining post-selection techniques, via optical and microwave spectroscopy of suitably strained SiVs and nuclear spin registers, together with independent integrability of the SiVs host into optical cavities[40–42], paves the way for future quantum networking experiments.

Cavity-assisted reflection/transmission[43,44] or emission-based approaches[5,45,46] could generate memory-photon entanglement which can then be distributed or frequency-converted into a common telecom band, enabling long-range quantum communication and measurement-based quantum computation[47–50]. During optical excitation and due to the high strain in the optical ground and excited state, we expect the quasi-free electron spin to experience less optically-induced memory dephasing, which was recently reported for a moderately-strained SnV center[15]. Beyond quantum networking experiments, integrating the highly coherent quasi-free electron spin into optomechanical systems, which typically exhibit coupling strengths in the few kHz regime, opens up new interesting physics toward transduction of quantum information[51].

## Methods
### Sample
We are using the same sample as well as the same nanodiamond as in ref. 17. The nanodiaomonds are grown in a high pressure (8 GPa) and high temperature (1450°C) process, where the silicon is introduced during growth. We estimate the size of the nanodiamond to be on the order of 100's of nm. Further details can be found in refs. 40,41,52.

The nanodiamonds are dispersed on a sapphire substrate with a 200 nm thick gold coplanar waveguide (CPW) for microwave supply. The nanodiamond is situated in the 10 μm gap of a 50 Ω CPW.

### Experimental setup
The sapphire substrate is placed on a custom copper cold finger inside a flow cryostat (Janis ST-500). The static magnetic field is applied with four neodymium permanent magnets in Hallbach configuration buried in the cold finger. We use a homebuilt 4-f confocal microscope for optical excitation and fluorescence collection, with a high NA objektiv (0.95NA 50x Olympus MPLAPON) at room temperature inside the vacuum chamber of the cryostat. Resonant excitation is done with a Ti:Sapphire laser (Sirah), pulsed with an acousto-optical modulator (AOM, G&H 3350–199). Resonantly excited photons from the phonon sideband are filtered with a bandpass filter, detected with an avalanche photo diode (Excelitas SPCM-AQRH-14-FC) and time tagged with a TimeTagger Ultra (Swabian Instruments). For coherent population trapping (CPT) experiments we modulate the laser with an electro-optical modulator (EOM, JENOPTIK AM705) locked into its transmission maximum (Toptica DLC Pro).

All electrical control signals (9.415 GHz for electron spin (MW), 2.9–4.2 MHz nuclear spins (RF) and 350 MHz AOM) are synthesized with one analog channel of an arbitrary waveform generator (Keysight M8195A) and split using diplexers (Mini-Circuits ZDSS-3G4G-S+ and ZDPLX-2150-S+). To increase the AWG's 8bit dynamic range for low-power electron spin control, we use a digital step attenuator (Analog Devices EVAL-ADRF5700) before amplifying the signal (Mini-Circuits ZVE-3W-183+). The RF signal is amplified (Mini-Circuits LZY-22+) and then combined with the MW before the cryostat using another diplexer (Mini-Circuits ZDSS-3G4G-S+). For the CPT experiments the microwave is synthesized with the AWG, amplified (Mini-Circuits ZVE-3W-83+) and combined with the lock output using a bias-tee (Mini-Circuits ZX85-12G-S+).

To set the attenuation level of the step attenuator, we use a digital channel of the AWG to send a number of pulses encoding the attenuation level. These are read with an FPGA (AMD Xilinx Zynq 7010, Red Pitaya STEMlab 125-14), which then sets the corresponding bits on the step attenuator. The counting of the photons for live feedback is done with the same FPGA. The active feedback is realized with a switch (Minicircuits ZASWA-2-50DRA+) controlled by the FPGA, which is deciding after a photon counting window whether a consecutive nuclear spin inverting RF-pulse is passed or dumped. The setup is orchestrated using Qudi.[53]

### Data processing and fitting
In each measurement, we extract the electron spin's population from the amplitude $a$ and offset $c$ of an exponential fit to a spin-pumping laser-pulse. If not indicated otherwise, the vertical error bars in the measurement data represent one standard deviation of exponential fit

**Table 2 | The parallel hyperfine coupling strengths $A_\parallel^i$, initialization fidelity $F_e$, Rabi period $T_{2\pi} = 2\pi/\Omega_{R,e}$ and inter-pulse spacing $\tau$ are fixed parameters in the fit of the XY($\tau$, $N$) measurements from the main text Fig. 2d to extract the Larmor frequency $\omega_{L,n}$ and the perpendicular hyperfine components $A_\perp^i$**

| $F_e(\tau)$ | $\tau$ µs | $T_{2\pi}$(ns) | $A_\parallel^i/2\pi$(kHz) | $\omega_{L,n}/2\pi$(kHz) | $A_\perp^i/2\pi$(kHz) |
|---|---|---|---|---|---|
| 0.8322 | 8.391 | 228 | 1194 kHz | 3.582 518(11) MHz | 233.19(81) |
| 0.8347 | 8.392 | | 420 kHz | | 147.7(15) |
| 0.8405 | 8.393 | | 121 kHz | | 75.5(35) |
| 0.8523 | 8.394 | | 34 kHz | | 46.8(44) |
| 0.8528 | 8.395 | | | | |
| 0.8402 | 8.396 | | | | |
| 0.8341 | 8.397 | | | | |
| 0.847 | 8.398 | | | | |
| 0.8302 | 8.399 | | | | |
| 0.8129 | 8.400 | | | | |
| 0.8089 | 8.401 | | | | |
| 0.845 | 8.402 | | | | |
| 0.875 | 8.403 | | | | |
| 0.8419 | 8.404 | | | | |

parameters. We extract the initialization fidelity in each measurement by applying a laser pulse after inverting the electron spin population with an unconditional $\pi$-pulse, leading to a maximal achievable contrast of $\langle\sigma_z\rangle = \langle\uparrow\rangle - \langle\downarrow\rangle = 2F_e - 1 = 0.72(2)$, where $F_e = a/(a+c)$[17]. We reference each readout pulse to the maximally achievable readout contrast. Errors indicated in brackets are one standard deviation of the respective fit parameter. All display items are created using Matplotlib.[54]

## Numerical model and system parameters

To evaluate the experimental data we are using a numerical model implemented with QuTIP describing the spin dynamics of our system[55,56]. The same model has been used in ref. 17 and we are only extending the number of nuclear spins we are taking into account. We model the system composed of four nuclear spins and one electron spin with the following Hamiltonian ($\hbar = 1$):

$$\widehat{H} = \frac{\omega_{L,e}}{2}\widehat{\sigma}_z^e + \sum_{i=1}^{N}\frac{\omega_{L,n}}{2}\widehat{\sigma}_{z,i}^n + \widehat{\sigma}_z^e\left(\frac{A_\parallel^i}{4}\widehat{\sigma}_{z,i}^n + \frac{A_\perp^i}{4}\widehat{\sigma}_{x,i}^n\right), \quad (1)$$

with the electron/nuclear Larmor frequencies $\omega_{L,e/n}$, hyperfine-coupling strengths $A_\parallel^i$ and $A_\perp^i$ and $N$ the number of nuclear spins, see Table 2. $\widehat{\sigma}_i$ are Pauli operators. All operators are understood to be of the right dimension, i.e. with additional identity operators $\mathbb{I}_2$. For example $\widehat{\sigma}_z^e \equiv \widehat{\sigma}_z \otimes \mathbb{I}_2 \otimes \mathbb{I}_2 \otimes \mathbb{I}_2 \otimes \mathbb{I}_2$.

Using eq. (1) we construct a suitable decoherence-free rotation $\mathcal{T}\exp(-i\int_0^t(\widehat{H}+\widehat{H}_D(t'))dt')$, where $\mathcal{T}$ is a time-ordering operator and for example

$$\widehat{H}_D(t) = \frac{\Omega_{R,e}(t)}{2}(\cos\phi(t)\widehat{\sigma}_x^e + \sin\phi(t)\widehat{\sigma}_y^e) \quad (2)$$

is a MW driving term on the electron spin with Rabi frequency $\Omega_{R,e}(t)$ and phase $\phi(t)$.

In the case of the dynamical decoupling measurements from Fig. 2 of the main text, rectangular-shaped pulses are used, centered in the nuclear spin spectrum, such that $\Omega_{R,e}(t) = const. = 2\pi/T_{2\pi}$ and

$\phi(t) = \omega_{L,e}t + \phi_0$. We then concatenate blocks of the form

$$\widehat{U}_{DD}(\tau) = [(\widehat{U}_0(\tau/2)\widehat{R}_x^N\widehat{U}_0(\tau/2)][(\widehat{U}_0(\tau/2)\widehat{R}_y^{N-1}\widehat{U}_0(\tau/2)]$$
$$\dots[(\widehat{U}_0(\tau/2)\widehat{R}_y^2\widehat{U}_0(\tau/2)][(\widehat{U}_0(\tau/2)\widehat{R}_x^1\widehat{U}_0(\tau/2)] \quad (3)$$

$$\widehat{U}_0(\tau/2) = \exp(-i\widehat{H}\tau/2) \quad (4)$$

$$\widehat{R}_{x,y} = \exp(-i(\widehat{H}+\widehat{H}_{D,\phi_0=0,\pi/2})T_{2\pi}/2) \quad (5)$$

to get the evolution $\widehat{U}_{DD}$ after $N$ blocks as a function of the inter-pulse spacing $\tau$ between two consecutive $\pi$-pulses. Note that $\widehat{H}$ is transformed into a frame rotating at $\omega_{L,e}$ such that the driving terms become time-independent.

We parametrize the initial density matrix by

$$\rho_0 = (F_e|\downarrow\rangle\langle\downarrow| + (1-F_e)|\uparrow\rangle\langle\uparrow|) \otimes \frac{\mathbb{I}_2}{2} \otimes \frac{\mathbb{I}_2}{2} \otimes \frac{\mathbb{I}_2}{2} \otimes \frac{\mathbb{I}_2}{2} \quad (6)$$

with electron initialization fidelity $F_e$ and statistical mixture on the nuclear spins.

In order to empirically account for loss of coherence we multiply off-diagonal terms of the electron spin's evolved density matrix $\rho_e(\tau) = tr_{n_i}[\widehat{U}_{DD}(\tau)\rho_0\widehat{U}_{DD}(\tau)^\dagger]$ with a function $\exp(-\tau/\tau_c)^\beta$ with $\tau_c = \tau_c^0 N^{\chi-1}$.

Specifically, for the 2D XY($\tau$, $N$) measurements from the main text, Fig. 2d, we construct a function with fixed $A_\parallel^i$ s calculated from the resonant RF frequencies, experimental parameters $F_e$, $T_{2\pi}$, $\tau$, $N$ and variable input parameters ($\omega_{L,n}$, $\tau_c^0$, $\beta$, $\chi$, $A_\perp^i$) and fit the dataset to the model with a least-square algorithm. In each measurement the number of decoupling pulses $N$ is swept from 20 to 380 in steps of 20. Table 2 shows the extracted hyperfine parameters, extracted Larmor frequency $\omega_{L,n}$ as well as the necessary experimental parameters, inter-pulse spacings $\tau$ and the electron spin initialization fidelity $F_e$ extracted from a reference-pulse in each measurement run.

These parameters are then used for the simulations of Fig. 3.

## Data availability

The authors declare that all data supporting the results and conclusions are presented in the main text and Supplementary Information. The raw data is stored on institutional servers and are available upon request from the corresponding author.

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

## Acknowledgements

We thank Matthias Müller for helpful discussion and simulations on using shaped sinc-pulses for enhanced sensitivity. We thank Guido van de Stolpe and Philipp Neumann for their fruitful input regarding nuclear spin spectroscopy. We thank V.A. Davydov and V. Agafonov for their initial contribution to the nanodiamond material. We thank the Ulm Center for Nanotechnology and Quantum Material for the metal deposition. The project was funded by the German Federal Ministry of

Research, Technology and Space (BMFTR) in the project QR.N (16KIS2208) (AK, MK, DO) and by the European Union Program QuantERA in the project SensExtreme (499192368)(AK, MK).

## Author contributions
A.T. prepared the sample. M.K. and A.T. set-up and conducted the experiments. M.K. developed the theoretical models and evaluated the data together with A.T. D.O. implemented the FPGA logic. M.K. wrote the manuscript together with A.T. The manuscript was discussed with M.K., A.T. and A.K. AK supervised the project. M.K. and A.T. contributed equally to this work.

## Funding

## Competing interests
The authors declare no competing interests.
