## [Transparent Peer Review file · Nature Communications]

Bipartite entanglement in a nuclear spin register mediated by a quasi-free electron spin

Corresponding Author: Professor Alexander Kubanek

Version 0:

Reviewer comments:

Reviewer #1

(Remarks to the Author)

The manuscript from M. Klotz et al. demonstrated two different ways of utilizing carbon nuclear spins as quantum registers using nearby silicon vacancy centers, namely, dynamical decoupling with varying time delay, and individual nuclear spin control using sinc-shaped RF pulses to provide selectivity. In my opinion, the overall demonstration is scientifically correct. However, the information flow of the manuscript is sometimes confusing, and the discussion of experimental results often becomes too technical without explanations of terms, assumptions, and results. This could make the manuscript too technical and complex to follow for the majority of the readers of Nature Communications. In addition, the manuscript requires structural improvements to emphasize its novelty and impact, both of which are needed for Nature Communications. My detailed comments are listed below:

1. In the abstract, the authors stated that the electron spin lifetime improvements could improve the sensing of nuclear-nuclear couplings to a few hertz. Unfortunately, I couldn't find anywhere in the rest of the manuscript that echoes this point.
2. In the abstract, the authors appraised the conditional phase gate method over dynamical decoupling, while in the dynamical decoupling section, it was noted that "our approach relies on dynamical decoupling...". It is unclear to me whether the author prefers this method or not. Since applying DD to manipulate nuclear spins is a well-established method, I am unsure why authors should devote a complete section and a figure if it is not the primary focus of the manuscript.
3. The authors would like to advertise the system as a potential candidate for quantum registers for quantum networks. I believe that additional discussions on why this particular system is a good candidate are necessary for discussing the impact of the technique. For instance, nano-diamonds usually come with random spatial location, color center frequency, strains (the yield of ultra-high strain centers is particularly important for the demonstration here), number of nuclear spins, and coherence, all of which will impact the scalability of the platform.
4. In the section "spin characterization of a quasi-free electron", it is unclear how much the static B field misalignment is. This is important in evaluating the cyclicity of the SiV. In fact, I am not sure why the discussion of cyclicity is here, as it seems to be weakly linked to the rest of the section.
5. In Figure 2 (a), it is confusing to label the parallel hyperfine coupling in the figure, partly because the dots are connected with lines without fitting, and it is not apparent where the peaks are. Perhaps the authors could move Equation 2 from Supplementary Note 5 here and discuss the model, measurement procedure, and Fourier transform, then label the 8 peaks in the figure so readers can understand the treatment more smoothly. In addition, the Fourier transform can be the lower panel while the Ramsey can be the top panel.
6. In the section "spin characterization of a quasi-free electron", the authors stated that free electrons are the dominant decoherence source of the color center. More discussion on the origin of the free electrons is preferred.
7. The fonts for most figures are too small for printed A4 paper, especially pulse sequences like Fig 3 (a), (e), Fig 5 (c), (e), (g), Fig 7 (b).

8. In the section "Dynamical decoupled nuclear spin detection and control", the authors mentioned the electron spin coherence under XY-N DD in a confusing way. It's not clear to me (1) why the authors mention this information here, while CPMG was discussed in the last section, and (2) why this information is placed after the nuclear spin discussion without any explanation. It seems detached from the rest of the material.

9. In Figure 3 (b), what are the fitting curves? In (f), it's unclear to me which nuclear spin curve is which due to almost identical colors.

10. In the section "direct nuclear spin spectroscopy", perhaps the authors can elaborate on why low-power MW pulses with slow Rabi oscillations are preferred, and why a longer T_2 _Rabi is observed for the general audience's reference.

11. In the section "Radio frequency nuclear spin control", the authors said that the sinc function is truncated at the second zero. What is the frequency profile of such a pulse? How different would this be from the ideal spectral shape? I am not sure how quantitative Figure 5 (b) is in terms of the pulses in the frequency domain.

12. I noticed a Rabi rate discrepancy in Figure 5 (d) n1, while the differences are not apparent for other nuclear spins. What is the possible cause?

13. In Figure 6 (b), the population histogram does not seem to be normalized. Some explanations are preferred.

14. I am not sure how Figure 7 (a) helps me understand the geometric phase gate operations. Perhaps some quantitative results on the detuning and the Rabi rate for different nuclear spin states can help evaluate this geometric phase in different conditions.

15. In Figure 7 (b), what is the big $\pi_x/2$ and $\pi_{x,y}/2$ gate? Are they two-qubit gates or two single-qubit gates?

16. In supplementary note 5, I think the total decoherence-free rotation cannot be written in such a simple form unless the Rabi rate is a constant with a simple linear $\phi(t)$. Instead, a time-ordered exponential with the integral of the Hamiltonian should be used.

Reviewer #2

(Remarks to the Author)

Thanks to the high strain property of SiV sample, and the low noise experimental setup, the authors are able to directly drive the nuclear registers using radio wave, instead of via dynamical decoupling which is limited by electron spin coherence time. This pathway enables the author to characterise the nuclear spins more thoroughly, and demonstrate Bell state entanglement.

While my overall impression of the manuscript is positive, I do have a few minor suggestions for the authors to improve it.

1. When reading through the abstract and Introduction, I was confusing by how many nuclear spins are there, as it ranges between two and four, eg. "entanglement of two ^{13}C nuclear spins", "directly detect and control up to four coupled nuclear spins", " 2π - rotation of the electron spin conditioned on three nuclear spins". Fig 1 is not referenced by the main text, but it is actually useful to the user when reading through the Introduction, so I recommend the author to reference the figure with qubit registered n1-4 labelled.

2. For each of the [SI], can the author to a specific section or figure within the document?

3. In page 3, '...entanglement of that nuclear spin with the electron spin leads to a loss of coherence...', I recommend modifying it to clarify which spin is losing coherence, eg. '... loss of electron coherence....'

4. Please indicate which nuclear spin Fig 3(c) refers to.

5. In page 3 of main text, the author showed a set of 4 numbers $A^i_{\text{perp}}/2\pi$, implying the dynamically decoupling experiment was performed on nuclear spins n1-n4. However, in each of the sub-figure in Fig. 3, the authors chose various number of qubits data to present, with no consistency. Eg. Fig 3(c) & (g) only shows n1, fig 3 (a),(b)&(e) shows both n1 & n2, fig 3(f) shows n1-n3. This makes the reader hard to keep track of which qubit the experiment is performed on. I recommend the authors to at least push the n3 data in fig 3(f) to supplementary, and add the equivalent fig 3 of qubits n3 and n4, if available, in the supplementary section.

6. Page 4, off-resonance DD:

a. By nature, the DD measurement performed here in Fig 3d is very similar to the the CPMG experiment above, the author should comment on the two numbers

b. T_2 of 2.31ms is on the same order of magnitude as the oscillation frequency in right of fig 3 b), which is concerning. Can the author points out this (and any other limitation) arise from the T_2 time?

7. In page 4, τ_{init} was used without being defined. Is it equivalent to τ in Fig 3(a)?

8. The manuscript indicates individual initialization is achieved at $\tau_{\text{init},1} = 8.3915 \mu\text{s}$, as indicated in Fig 3f. Is the criteria maximal $\langle \rho_z \rangle$? If yes, then why is 8.3915us chosen over 8.3955us?

9. In Fig 5b, the boundaries of the shade regions are pixelated. Is it an artefact or does it indicates the frequency resolution of the sinc function?

10. In page 6, why a pulse of 150kHz is chosen? given that the smallest hyperfine energy splitting is $\sim 141\text{kHz}$ (n3), a pulse of 150kHz doesn't seem to have sufficient spectral resolution to distinguish spin up and down of n3.

11. In page 7, has the authors attempt prepare other 3 Bell states? If so, what is the fidelity? Depends on the locality of the noise source, parallel Bell states might yield higher state fidelity.

After a minor revision stated above, I would like to recommend an acceptance of this manuscript to Nature communications.

Reviewer #3

(Remarks to the Author)

The manuscript demonstrates full control of a three-qubit ^{13}C nuclear spin register coupled to a highly strained SiV center in nanodiamond. By exploiting the quasi-free electron spin regime under extreme strain, the authors achieve initialization, two-qubit operations, and single-shot readout of all coupled nuclear spins — essentially transplanting well-established NV-center techniques to the SiV platform.

The paper is technically complete, well written, and the experiments are carefully executed. The work overcomes the long-standing limitation of SiV systems, where spin control had lagged behind their excellent optical interface properties. It therefore constitutes a solid step toward implementing local quantum registers for quantum network nodes. However, the conceptual advance is moderate: the results mainly represent an incremental integration of known control methods rather than a fundamentally new paradigm. I recommend major revision before the manuscript can be considered for publication.

Comments

1. On page 1, the authors state that “local entanglement of highly coherent ^{13}C nuclear spins, necessary for error detection and correction, has so far only been demonstrated with spin-1 NV.” This raises the question of whether the spin-1 itself is essential for realizing such operations, and, if the quasi-free electron spin plays a distinct role, what specific advantage it provides beyond simple orbital decoupling.

2. The identification of a fourth ^{13}C spin seems tentative. Given the dense nuclear environment, overlapping or near-degenerate hyperfine couplings may yield similar spectral features. Please provide additional supporting evidence to substantiate the claim that the fourth spin is unambiguously resolved.

3. The demonstrated register relies on three strongly coupled ^{13}C nuclear spins in a specific nanodiamond, which is impressive but highly sample-dependent. The authors should discuss possible routes to scalability. It would also be useful if they could comment on how their strongly coupled register compares conceptually with approaches based on more weakly coupled or isotopically engineered nuclear spins that have been explored in the field.

Version 1:

Reviewer comments:

Reviewer #1

(Remarks to the Author)

I think the revised manuscript has clearly improved over the last version, with much less confusion and enriched content. I recommend publication to Nature Communications.

Reviewer #2

(Remarks to the Author)

I appreciate the authors taking the time to address all of my previous concerns point by point, and make major changes to the manuscript and the supplementary materials to accommodate my comments and suggestion.

I have noted my response in the attached file, under each of the dot points the authors have responded to.

With the answer and modification the authors have provided, my overall impression of the manuscript remains positive, and I would still recommend the manuscript to be published in Nature Communications

Reviewer #3

(Remarks to the Author)

I thank the authors for their responses and for the revisions made to the manuscript and the Supplementary Information. The authors have addressed my previous comments by adding further analyses and simulations, which helped clarify several points.

The revised Introduction and Conclusion make it clearer why the authors focus on the quasi-free electron regime. Compared to the original version, the manuscript now better explains how this work differs from a straightforward application of NV-center techniques to the SiV system.

1. Conceptual aspects

The authors' explanation is clearer than before. The use of low-power microwave sensing provides a reasonable way to mitigate sensitivity limitations in spin-1/2 systems. In addition, the robustness enabled by strong strain—such as operation at 4 K and the use of permanent magnets—can be seen as an advantage of the SiV platform in realistic quantum-network

settings. Emphasizing these points improves the presentation of the work.

2. Identification of the nuclear spin register

The additional nuclear Rabi spectroscopy data and the analysis in Supplementary Note 10 address my concerns regarding the identification of the fourth nuclear spin. The agreement between the measured RF shifts and the expected values supports this interpretation.

3. Scalability

The Monte Carlo simulations added in Supplementary Note 3 provide a clearer picture of scalability, showing that the probability of obtaining a three-qubit register at natural abundance is low (0.05%). While this remains a limitation, I appreciate that the authors state this explicitly and discuss possible directions for improvement, such as post-selection and nuclear-spin bath cooling.

In summary, the manuscript has improved through clearer explanations, additional technical detail, and a more explicit discussion of limitations. While I remain cautious, the revised version is more convincing than the original and may be suitable for publication in Nature Communications.

REVIEWER COMMENTS

General Notes:

- We noticed a minor mistake in the data evaluation regarding error propagation of normalized readout amplitudes. This insignificantly changes the error of the register state preparation as well as the error of the correlators from $(-0.645(27), 0.608(27), 0.502(25))$ to $(-0.645(39), 0.608(38), 0.502(38))$ and thus the error of the fidelity Bell-states from $0.689(11)$ to $0.689(17)$.
We modified the error values accordingly.
- We corrected spelling errors and inserted missing factors of 2π
- We minorly rephrased additional words and small parts of sentences to help the information flow
- We reworked the nuclear spin visualizations

Reviewer #1 (Remarks to the Author):

The manuscript from M. Klotz et al. demonstrated two different ways of utilizing carbon nuclear spins as quantum registers using nearby silicon vacancy centers, namely, dynamical decoupling with varying time delay, and individual nuclear spin control using sinc-shaped RF pulses to provide selectivity. In my opinion, the overall demonstration is scientifically correct. However, the information flow of the manuscript is sometimes confusing, and the discussion of experimental results often becomes too technical without explanations of terms, assumptions, and results. This could make the manuscript too technical and complex to follow for the majority of the readers of Nature Communications. In addition, the manuscript requires structural improvements to emphasize its novelty and impact, both of which are needed for Nature Communications. My detailed comments are listed below:

Before answering point by point, we would like to thank the Reviewers insightful concerns and suggestions, which helped us improve the manuscript's quality.

1. In the abstract, the authors stated that the electron spin lifetime improvements could improve the sensing of nuclear-nuclear couplings to a few hertz. Unfortunately, I couldn't find anywhere in the rest of the manuscript that echoes this point.

We thank the author for finding this inconsistency.

We addressed it now in the manuscript by adding the following sentence:

“Due to the electron spin's $T_{1,e}$ the phase accumulation of the sensor nuclear spin gets randomized, limiting sensitivity to few Hz.”

2. In the abstract, the authors appraised the conditional phase gate method over dynamical decoupling, while in the dynamical decoupling section, it was noted that "our approach relies on dynamical

decoupling...". It is unclear to me whether the author prefers this method or not. Since applying DD to manipulate nuclear spins is a well-established method, I am unsure why authors should devote a complete session and a figure if it is not the primary focus of the manuscript.

We would like to make the Reviewer aware that we wrote "One approach" instead of "our approach". DD-nuclear spin control is limited in sensitivity to $(A_{\perp}/\omega_{L,n})^2$ when detecting nuclear spins due to the spin-1/2 nature of our defect centers. However, DD allows detection of nuclear spins with high A_{\perp} and low A_{\parallel} , which are not detectable with the approach relying on low-power microwave pulses.

We changed in the manuscript:

"One well-established method to characterize the nuclear spin environment ..."

3. The authors would like to advertise the system as a potential candidate for quantum registers for quantum networks. I believe that additional discussions on why this particular system is a good candidate are necessary for discussing the impact of the technique. For instance, nano-diamonds usually come with random spatial location, color center frequency, strains (the yield of ultra-high strain centers is particularly important for the demonstration here), number of nuclear spins, and coherence, all of which will impact the scalability of the platform.

We would like to emphasize that the presented results are a first demonstration of ^{13}C - ^{13}C entanglement in a quantum register coupled to a group-IV defect in diamond – and to the best of our knowledge are the first demonstration of these results with an optically accessible spin-1/2 system. The presented methods are applicable in general for spin-1/2 systems. Furthermore, we demonstrate that the problems mentioned by the reviewer do not hold in general for SiVs hosted in nanodiamonds. In fact, we demonstrate that our highly-strained SiV, albeit hosted in a nanodiamond, can be used to study local entanglement and in future experiments will serve as a platform for quantum network experiments.

We do not offer statistics to prove the system's scalability, but would still like to address the raised concerns as we envision a scalable system in the future. We would like to emphasize, that we chose nanodiamonds as host material, to be able to post-select suitable SiVs and later integrate their host using an AFM-based pick&place technique into photonic structures necessary for quantum networking applications.

We would like to address the raised points point by point:

- *Spatial location: AFM nanomanipulation allows transfer and alignment of NDs with resolution of the AFM. The coupling of SiVs to optical infrastructure can be controlled with higher precision than the uncertainty in estimating the position of the SiV [10.1021/acsp Photonics.3c01559]*
- *Color center frequency: Fast resonant confocal scans allow identification of indistinguishable emitters in different nanodiamond [10.1515/nanoph-2023-0379] More importantly, in a future network, up-conversion to the telecom-band further relaxes indistinguishability demands.*
- *Strain: Fast optical spectroscopy allows measurement of strain/ground-state splitting to identify suitable SiV. Also, the yield of strained defects can be improved by for example laser induced vacancy migration or material combinations [see citation 24 and 25].*

- *Coherence: Direct consequence of the high strain is a suppression of phonon-induced decoherence. Different noise sources have to be excluded by coherence measurements of suitable SiV centers*
- *Number of nuclear spins: using nuclear spins as resource is by nature probabilistic and poses the largest scalability concerns. We performed Monte-Carlo simulations to estimate the probability of finding a SiV with suitable register size, see new Supplementary Note 3. From there we conclude that cooling of the nuclear spin bath and a consequentially improved sensitivity could provide a suitable solution to improve the yield of usable nuclear spin registers.*

Together with the suggestions of Reviewer 3 regarding advantages of the quasi-free electron spin, we now reworked the “Conclusion and outlook” and “Introduction” section of the manuscript to highlight more precisely the conceptual and technical advancements and impact of our findings.

4. In the section "spin characterization of a quasi-free electron", it is unclear how much the static B field misalignment is. This is important in evaluating the cyclicity of the SiV. In fact, I am not sure why the discussion of cyclicity is here, as it seems to be weakly linked to the rest of the section.

Inline with the Reviewers suggestion we rephrased the part containing the cyclicity and now write

“Although we measured a relaxation time of $T_{1,e} = 0.296(85)s$ and a cyclicity of $\eta = 2020(380)$, see Suppl. Fig.1, we are unable to profit from single-shot readout of the electron spin to increase the readout fidelity, due to a photon collection efficiency on the order 0.1%.”

Indeed, for low-strained SiVs the magnetic field alignment is the key parameter influencing the cyclicity. Due to the high strain however, we can retain a high cyclicity even under moderate B-Field misalignment as has been shown in [16, 18]. This allows us to use permanent magnets which however does not allow us to determine the exact magnetic field alignment.

5. In Figure 2 (a), it is confusing to label the parallel hyperfine coupling in the figure, partly because the dots are connected with lines without fitting, and it is not apparent where the peaks are. Perhaps the authors could move Equation 2 from Supplementary Note 5 here and discuss the model, measurement procedure, and Fourier transform, then label the 8 peaks in the figure so readers can understand the treatment more smoothly. In addition, the Fourier transform can be the lower panel while the Ramsey can be the top panel.

We thank the reviewer for pointing out the confusing labeling in Fig. 2a. We now changed the figure according to the Reviewers suggestion, which now more clearly shows the measurement and data-processing/fitting procedure. Since the model was not directly involved in Fig. 2a and to avoid further extending the Fig. caption, we didn't move Equation 2.

6. In the section "spin characterization of a quasi-free electron", the authors stated that free electrons are the dominant decoherence source of the color center. More discussion on the origin of the free electrons is preferred.

In-line with the Reviewer's suggestion, we now give more context of the potential source of free electron by modifying the following sentence:

“We attribute the dominant noise source to a bath of free electrons potentially originating from surface defects of the host or other defect center in the vicinity [17].”

7. The fonts for most figures are too small for printed A4 paper, especially pulse sequences like Fig 3 (a), (e), Fig 5 (c), (e), (g), Fig 7 (b).

We agree with the reviewer and reworked the fonts in all figures to be more readable.

8. In the section "Dynamical decoupled nuclear spin detection and control", the authors mentioned the electron spin coherence under XY-N DD in a confusing way. It's not clear to me (1) why the authors mention this information here, while CPMG was discussed in the last section, and (2) why this information is placed after the nuclear spin discussion without any explanation. It seems detached from the rest of the material.

We thank the Reviewer for raising this point. The reason why we mentioned the coherence time under XY-N DD is that it's a different type of measurement. We fixed τ and swept N , compared to the set of CPMG measurements in Fig. 2b, where we swept τ for specific fixed N s and extracted a $T_{2,e}$ for each measurement separately to extract the scaling factor.

The XY-N DD coherence measurement presented in Fig.3d(now f) probes the environments noise spectrum, off-resonant with the nuclear spin bath. Additionally, this sets an upper bound for our sensitivity of detecting nuclear spins, as can be seen in Fig. 3b.

We now reworked the section to make the difference more understandable, and moved the discussion of the coherence towards the end of the section.

“In order to probe the electron spin's coherence under XY-N DD we choose $\tau = 8.43 \mu\text{s}$, off-resonant with a multiple of the nuclear spins precession period, sweep N and observe an exponential decay in coherence within $T_{2,e}^{XY} = 2.31(14)\text{ms}$, shown in Fig.3f. The measured $T_{2,e}^{XY}$ poses an ultimate limit to our sensitivity of detecting nuclear spins with DD and limits indirect nuclear spin control gates, as can be seen by the exponential decaying oscillations in Fig.3b. It is also worth noting that $T_{2,e}^{XY}$ differs from an extrapolation of the CPMG measurements, which yields $T_{2,e}^{CPMG,274} = T_{2,e}^{CPMG,32} \cdot (274/32)^{\chi} \approx 3.9\text{ms}$. The discrepancy is attributed to the specific noise characteristics at $\tau = 8.43 \mu\text{s}$ or pulse errors at large $N \gg 32$.”

9. In Figure 3 (b), what are the fitting curves? In (f), it's unclear to me which nuclear spin curve is which due to almost identical colors.

Regarding Fig. 3b we used the fitted numerical model from the 2D dataset shown in c. In order to clarify this more clearly, we modified the figure caption of Fig.3b regarding the subfigure b:

“Dash-dotted lines are simulations using the numerical model fitted with the 2D dataset.”

We also introduced white dashed lines in Fig.3c to highlight the data shown in Fig.3b. Additionally, we moved Fig. 3f to the Supplemental Note 7 to divert less attraction from the main results and give more context.

10. In the section "direct nuclear spin spectroscopy", perhaps the authors can elaborate on why low-power MW pulses with slow Rabi oscillations are preferred, and why a longer $T2_{\text{Rabi}}$ is observed for the general audience's reference.

Low power microwave pulses are used to selectively address nuclear spin split electron spin transitions, since the Rabi frequency sets the lower bound for the ability to discriminate individual resonances which are split by A_{\parallel} .

The reason for an extension in electron spin coherence during driving compared to for example a Ramsey measurement can be understood in a dressed state picture, where the energy barrier of the dressed states is given by the Rabi frequency. As long as the dephasing noise has a lower frequency or magnitude than the Rabi frequency, the noise merely alters the dressed states. The Reviewer is pointed to <https://www.frontiersin.org/journals/quantum-science-and-technology/articles/10.3389/frqst.2023.1228208/full> for a more detailed discussion.

We modified the phrases:

“[...] an order of magnitude longer than $T_{2,e}^*$, enabled by continuous DD of the electron spin [30].”

“We scan for various nuclear spin-dependent resonances sweeping the frequency ν_{MW} MW of a [...], setting the lower bound for the spectral resolution.”

11. In the section "Radio frequency nuclear spin control", the authors said that the sinc function is truncated at the second zero. What is the frequency profile of such a pulse? How different would this be from the ideal spectral shape? I am not sure how quantitative Figure 5 (b) is in terms of the pulses in the frequency domain.

The frequency profile of a truncated sinc-pulse, shown in Fig. 5a, is depicted in Fig. 5b. The deviation from an ideal shape is most prominent at the limited steepness outside the bandwidth as well as the non-flatness within the bandwidth. Using a higher-order truncation would improve the shape and hence spectral selectivity at the cost of longer MW pulses where decoherence might become problematic. Our motivation for Fig. 5b is to visualize the construction of the respective CNOTs in the spectral domain.

12. I noticed a Rabi rate discrepancy in Figure 5 (d) n1, while the differences are not apparent for other nuclear spins. What is the possible cause?

We thank the Reviewer for this detailed observation. L. Childress et al. observed similar results with NV centers and argued that the different hyperfine couplings between electron and nuclear spins lead to different electronic admixture to the magnetic moment of the respective nuclear spin which leads to enhanced Rabi frequency [10.1126/science.1131871]. A very recent work from J. Resch et al [10.48550/arXiv.2509.03354] also observed similar behavior on strongly coupled ^{13}C spin.

Our simulations in Fig.5 (d) take the Rabi frequency as a given parameter from a qualitative fit, i.e. the simple model is not reflecting this admixture but takes it explicitly into account.

To make the reader aware of this we modified:

“It is worth noting that we observe a difference in Rabi frequencies which is higher for larger hyperfine couplings, possibly due to an electron spin-induced modification of the nuclear spin’s magnetic moment [33], and not reflected in the numerical model, where we explicitly included the measured Rabi frequencies.”

13. In Figure 6 (b), the population histogram does not seem to be normalized. Some explanations are preferred.

We thank the Reviewer for pointing this out. Currently we plotted the amplitudes of each readout signal relative to our initial strong calibration pi-pulse. This includes readout errors, such that the summed

amplitudes is ~73% of the calibration amplitude, inline with the 150 kHz readout pulse fidelity. However, this is not in accordance with the y-Axis label "Population".

We corrected the measured values for the known readout infidelity to display as indicated the measured population.

14. I am not sure how Figure 7 (a) helps me understand the geometric phase gate operations. Perhaps some quantitative results on the detuning and the Rabi rate for different nuclear spin states can help evaluate this geometric phase in different conditions.

We agree with and thank the reviewer that most of the subfigure 7(a) is not containing much information to understand the 2π phase-gate. We modified the sketch and incorporate a resonant and detuned case and visualize the corresponding trajectories, solid angles and geometric phases.

15. In Figure 7 (b), what is the big $\pi_x/2$ and $\pi_{x,y}/2$ gate? Are they two-qubit gates or two single-qubit gates?

The single-qubit gates are indeed misleadingly illustrated. Both blocks are single-qubit gates. However, the larger block represents two single-qubit gates on both nuclear spins, similarly to the "Init" block where all three nuclear spins are initialized. We now depict the single-qubit gates with separate blocks to avoid confusion.

16. In supplementary note 5, I think the total decoherence-free rotation cannot be written in such a simple form unless the Rabi rate is a constant with a simple linear $\phi(t)$. Instead, a time-ordered exponential with the integral of the Hamiltonian should be used.

We thank the Reviewer for pointing out the incorrectly used simple form and now changed the Hamiltonian accordingly.

Reviewer #2 (Remarks to the Author):

Thanks to the high strain property of SiV sample, and the low noise experimental setup, the authors are able to directly drive the nuclear registers using radio wave, instead of via dynamical decoupling which is limited by electron spin coherence time. This pathway enables the author to characterise the nuclear spins more thoroughly, and demonstrate Bell state entanglement.

While my overall impression of the manuscript is positive, I do have a few minor suggestions for the authors to improve it.

Before answering point by point, we would like to express our appreciation of the Reviewers overall suggestions, since we think this helped the manuscript's quality.

1. When reading through the abstract and Introduction, I was confusing by how many nuclear spins are there, as it ranges between two and four, eg. "entanglement of two ^{13}C nuclear spins", "directly detect and control up to four coupled nuclear spins", " 2π - rotation of the electron spin conditioned on three nuclear spins". Fig 1 is not referenced by the main text, but it is actually useful to the user when reading through the Introduction, so I recommend the author to reference the figure with qubit registered n1-4 labelled.

We agree with the Reviewer and modified the introduction, inline with suggestions from Reviewer 1 and 3, to make it clearer that we are controlling a three-qubit nuclear spin register with an auxiliary fourth nuclear spin, which is not used within the register. Additionally, we are directly referencing Fig.1 in that context.

2. For each of the [SI], can the author to a specific section or figure within the document?

We are now referencing the [SI] more precisely with the specific section within the main manuscript.

3. In page 3, '..entanglement of that nuclear spin with the electron spin leads to a loss of coherence...', I recommend modifying it to clarify which spin is losing coherence, eg. '... loss of electron coherence....'

We thank the reviewer for the suggestion and modified the respective sentence accordingly.

"Under the right conditions, i.e. the right inter-pulse spacing τ and number of π -pulses N in the pulse sequence depicted in Fig.3a, entanglement of that nuclear spin with the electron spin leads to a loss of electron spin coherence [17,23]."

4. Please indicate which nuclear spin Fig 3(c) refers to.

Fig.3c is addressing all nuclear spins, especially oscillations of the two strongest nuclear spins are visible, highlighted now with white dashed lines. We added the nuclear spin illustrations to Fig.3b&e to make a proper visual connection between the plots.

5. In page 3 of main text, the author showed a set of 4 numbers $A^i_{\text{perp}}/2\pi$, implying the dynamically decoupling experiment was performed on nuclear spins n_1 - n_4 . However, in each of the sub-figure in Fig. 3, the authors chose various number of qubits data to present, with no consistency. Eg. Fig 3(c) &(g) only shows n_1 , fig 3 (a),(b)&(e) shows both n_1 & n_2 , fig 3(f) shows n_1 - n_3 . This makes the reader hard to keep track of which qubit the experiment is performed on. I recommend the authors to at least push the n_3 data in fig 3(f) to supplementary, and add the equivalent fig 3 of qubits n_3 and n_4 , if available, in the supplementary section.

We thank the Reviewer and in addition to the modifications/clarifications of the Reviewer's 4th comment we now pushed Fig3.f (old) to the Supplementary Note 7.

We extracted the set of four coupling components from the 2D-dataset presented in Fig.3c, where all coupled spins contribute. However, we could not individually resolve coherent oscillations of n_3 and n_4 as we were able to do for n_1 and n_2 in Fig.3b.

6. Page 4, off-resonance DD:

- a. By nature, the DD measurement performed here in Fig 3d is very similar to the the CPMG experiment above, the author should comment on the two numbers
- b. T_2 of 2.31ms is on the same order of magnitude as the oscillation frequency in right of fig 3 b), which is concerning. Can the author points out this (and any other limitation) arise from the T_2 time?

In the updated manuscript the labeling of Fig.3d is now Fig.3f.

a: We thank the reviewer for bringing up a potentially helpful comparison between the two measurements. We would like to mention that the CPMG measurements in Fig.2b is physically different from off resonant DD measurements in the following sense. The CPMG measurements on the one hand sweep the inter-pulse spacing, thus probing the surrounding noise spectrum at different frequencies (for a superset of order N). Given the scaling χ we can extrapolate $T_{2,e}^{\text{CPMG},274}$ for 274 decoupling pulses to $T_{2,e}^{\text{CPMG},274} = T_{2,e}^{\text{CPMG},32} \cdot (274/32)^\chi \approx 3.9\text{ms}$, i.e. higher than what we have measured in Fig.3d (now f).

The order sweep in Fig.3d (now f) on the other hand, probes the noise's width at a specific frequency component by keeping the inter-pulse spacing fixed and basically narrowing the filter-functions bandwidth. The discrepancy between CPMG and XY might thus be attributed to specific noise characteristics at $\tau = 8.43\mu\text{s}$ or remaining pulse errors which are not compensated at very large N . The latter might not have been visible in CPMG measurements. Both causes can be investigated in future works but are not the focusing here.

For clarification and deeper insights into the presented measurements, we now added an additional sentence with the comparison presented here since. Also, we moved the sentence to the end of the DD section, for a better flow of information. Inline with comment 8 of Reviewer #1 we added:

"In order to probe the electron spin's coherence under XY-N DD we choose $\tau = 8.43 \mu\text{s}$, off-resonant with a multiple of the nuclear spins precession period, sweep N and observe an exponential decay in coherence within $T_{2,e}^{\text{XY}} = 2.31(14)\text{ms}$, shown in Fig.3f. The measured $T_{2,e}^{\text{XY}}$ poses an ultimate limit to our sensitivity of detecting nuclear spins with DD and limits indirect nuclear spin control gates, as can be seen by the exponential decaying oscillations in Fig.3b. It is also worth noting that $T_{2,e}^{\text{XY}}$ differs from an

extrapolation of the CPMG measurements, which yields $T_{2,e}^{CPMG,274} = T_{2,e}^{CPMG,32} \cdot (274/32)^{\chi} \approx 3.9\text{ms}$. The discrepancy is attributed to the specific noise characteristics at $\tau = 8.43 \mu\text{s}$ or pulse errors at large $N \gg 32$."

7. In page 4, τ_{init} was used without being defined. Is it equivalent to τ in Fig 3(a)?

We appreciate the Reviewer's comment on a missing proper definition of τ_{init} , which was indirectly defined in the caption of Fig.3f (old). To improve clarity and in conjunction with the Reviewers suggestion 5, we now modified the manuscript and referenced a section to the Supplementary Figure 5:

"[...] requiring electron spin conditioned $\pi/2$ rotations of the target nuclear spin with a slightly off-resonant τ_{init} , see the sequence depicted in Fig.3d. We implement the corresponding sequence with $N = 24$ and sweep τ_{init} around the resonance, where individual initialization is achieved at $\tau_{init,1} = 8.3915 \mu\text{s}$, see Suppl. Fig. 5."

8. The manuscript indicates individual initialization is achieved at $\tau_{init,1} = 8.3915 \mu\text{s}$, as indicated in Fig 3f. Is the criteria maximal $\langle \rho_z \rangle$? If yes, then why is $8.3915\mu\text{s}$ chosen over $8.3955\mu\text{s}$?

We thank the Reviewer for this observation and would like to address it with the following argument. Since the nuclear spin's initialization and readout fidelity with this sequence is limited by the additional nuclear spins we are not able to differentiate whether any nuclear spin is better or worse initialized than the other. In order to independently measure the initialization fidelities, depicted in Supplementary Fig. 5, one would have to use a different measurement technique, see for example [18], Fig.3c.

We chose the resonance, because our numerical model predicts for this exact resonance polarization of only one nuclear spin, facilitating the interpretation of the measured Ramsey signal. We added in Supplementary Note 7 a discussion about the choice of τ_{init} .

9. In Fig 5b, the boundaries of the shade regions are pixelated. Is it an artefact or does it indicate the frequency resolution of the sinc function?

The pixelated-looking boundary is indeed due to the limited frequency resolution set by the truncated sinc function in the time-domain.

10. In page 6, why a pulse of 150kHz is chosen? given that the smallest hyperfine energy splitting is $\sim 141\text{kHz}$ (n_3), a pulse of 150kHz doesn't seem to have sufficient spectral resolution to distinguish spin up and down of n_3 .

We thank the Reviewer for bringing up this concern. In an ideal scenario, i.e. infinitely temporal extension of the pulse, the resonant 150 kHz sinc-pulse's spectral window would not cause spectral leakage into the next pair of resonances, making it a conditional pulse. This is due to the fact the resonances of n_3 are split by 141 kHz such that the pulse's bandwidth only accommodates roughly half the splitting, namely $150 \text{ kHz}/2=75 \text{ kHz} < 141 \text{ kHz}$. However, due to our truncation and the aforementioned frequency resolution there is some expected small spectral leakage which we simulated in the Supplementary Note 9, when considering the amplitude sweeps to implement CNOTs.

Reducing the bandwidth would've helped considering the mentioned argument but would then have caused issues with the next weaker coupled nuclear spins, which have to be addressed unconditionally. For this reason, we chose 150 kHz. Further experimental optimization of the gate fidelity regarding truncation and bandwidth or a different pulse-shape altogether could be an object of future experiments.

11. In page 7, has the authors attempt prepare other 3 Bell states? If so, what is the fidelity? Depends on the locality of the noise source, parallel Bell states might yield higher state fidelity.
After a minor revision stated above, I would like to recommend an acceptance of this manuscript to Nature communications.

We thank the Reviewer for bringing up this interesting insight. We indeed measured the other three Bell states. The one we are presenting currently in the main manuscript turned out to have the highest fidelity. For the data of the other Bell states with discussion, the Reviewer is referred to the new Supplemental Note 12.

Reviewer #3 (Remarks to the Author):

The manuscript demonstrates full control of a three-qubit ^{13}C nuclear spin register coupled to a highly strained SiV center in nanodiamond. By exploiting the quasi-free electron spin regime under extreme strain, the authors achieve initialization, two-qubit operations, and single-shot readout of all coupled nuclear spins — essentially transplanting well-established NV-center techniques to the SiV platform. The paper is technically complete, well written, and the experiments are carefully executed. The work overcomes the long-standing limitation of SiV systems, where spin control had lagged behind their excellent optical interface properties. It therefore constitutes a solid step toward implementing local quantum registers for quantum network nodes. However, the conceptual advance is moderate: the results mainly represent an incremental integration of known control methods rather than a fundamentally new paradigm. I recommend major revision before the manuscript can be considered for publication.

Comments

1. On page 1, the authors state that “local entanglement of highly coherent ^{13}C nuclear spins, necessary for error detection and correction, has so far only been demonstrated with spin-1 NV.” This raises the question of whether the spin-1 itself is essential for realizing such operations, and, if the quasi-free electron spin plays a distinct role, what specific advantage it provides beyond simple orbital decoupling.

We would like to thank the Reviewer for this comment. In the quoted sentence, we were referring to the diamond community, where local ^{13}C registers have so far only been demonstrated with spin-1 NV. Additionally, we would like to point the reviewer towards dopants in silicon, where nuclear spin entanglement via an electron spin-1/2 locally [10.1038/s41467-024-52795-4] and very recently non-locally have been demonstrated [10.1126/science.ady379].

However, we believe that optically accessible defect centers are necessary to achieve scaling of the qubit network by optical means. Although impressive progress is made with optically active defect centers in different host materials, such as Silicon Carbide, Silicon or rare earth ions in crystals and molecules, we would like to compare in the following the register capabilities of spin-1 NV, which is the benchmark in solid-state qubit registers, and spin-1/2 SiV in diamond.

The main advantage of spin-1 systems is the linear dependence of resonances of individual nuclear spins on the A_{\parallel} / ω_L hyper-fine coupling component in dynamically decoupled spectroscopy and control as compared to only second order dependence on A_{\perp} / ω_L for spin-1/2 [28]. While this can be mitigated by increasing the free precession intervals, as has been shown in our paper, where we detect two nuclear spin resonances in a DD spectrum with a spin-1/2 defect, the applicability is limited. One way around this is DD-RF-control, where coherence protection on the electron and direct driving of the nuclear spin can be used to detect coupling between them. The selectivity to identify nuclear spins by selective nuclear spin driving is the same for spin-1/2 and spin 1 systems, although spin-1 systems profit from a different phase build-up due to different average Larmor precession frequencies of the used qubit-states which increases selectivity, see [https://doi.org/10.1103/PRXQuantum.6.020309]. First experimental demonstrations on spin-1/2 SnV centers in diamond have shown access to two nuclear

spins, limited by the spectral selectivity of RF pulses, which is limited by the inter-pulse spacings between electron decoupling pulses, and therefore electron coherence.

We believe that a quasi-free electron can solve the coherence problem of group-IV defects. Due to spin and orbital decoupling, dephasing due to phonon absorption is mitigated, as we demonstrated by measuring >1ms coherence times at elevated temperatures in this paper.

Furthermore, the approach to detect and entangle nuclear spins relying on low-power MW pulses, demonstrated in this paper, is also linearly sensitive to parallel hyperfine couplings, A_{\parallel} . To be able to implement these gates we were relying on beneficial properties of the quasi-free electron. Due to less sensitivity to the alignment of the external field for a quasi-free electron, we can profit from He-cooled permanent magnets which supply a highly stable magnetic field. We observe highly stable electron spin transitions over months and several cooling cycles. We attribute this again to the strong orbital decoupling and hence less phonon susceptibility, even inside a nanometer-sized host which presented a challenge in nanocavities.

Our ability to measure Rabi frequencies down to few kHz to sense weakly coupled nuclear spins, see Supplementary Note 8, additionally opens up new physics for example when envisioning coupling these SiV containing nanodiamonds to superconducting MW or mechanical resonator structures which typically have low magnetic or mechanical coupling strength, respectively. This can be useful for transduction of quantum information from either nuclear or electron spins.

Thanks to the quasi-free electron spin we also expect less optically induced decoherence on nuclear spins due to the additional orbital decoupling in the optical excited state, such that we would expect a preferential absence of precession frequency/axis change of nuclear spins when exciting the optical dipole that would otherwise lead to dephasing during entanglement attempts with a photon. This is another object of future experiments.

In conclusion, we think that spin-1/2 systems, and especially our highly strained SiV with a quasi-free electron spin will be able to fulfill all requirements for a locally entangled and optically accessible spin register.

In order to highlight the capabilities of a quasi-free electron spin-1/2 more detailed, we now modified the "Introduction" and "Conclusion and outlook" section accordingly.

2. The identification of a fourth ^{13}C spin seems tentative. Given the dense nuclear environment, overlapping or near-degenerate hyperfine couplings may yield similar spectral features. Please provide additional supporting evidence to substantiate the claim that the fourth spin is unambiguously resolved.

We treat the fourth nuclear spin in the manuscript as an additional source of error which is at the limit of our detection capabilities. However, we can present three more arguments manifesting the claim of its presence

In case of near-degenerate couplings in the form of parallel couplings $A_{\parallel}^4 \approx A_{\parallel}^5$ which dominate the spectrum due to the large B field, we would suspect to observe a different pulsed ODMR signal on the electron spin, i.e. a strong peak in the center from the two combinations $\mp A_{\parallel}^4 \pm A_{\parallel}^5$ with two side-peaks which are at $\pm A_{\parallel}^4 \pm A_{\parallel}^5$.

The RF frequency during the nuclear Rabi measurements on the fourth nuclear spin, turned out to be close to 3.5639 MHz and 3.60143 MHz, for electron spin down/up respectively. This is in good agreement with an A_{\parallel} –dominated nuclear resonance shift of $A_{\parallel,\downarrow}^4/2 \approx 3.5825$ MHz – 3.5639 MHz ≈ 18.6 kHz and $A_{\parallel,\uparrow}^4/2 \approx 3.60143$ MHz – 3.5825 MHz ≈ 18.9 kHz.

When driving this RF resonance at $\Omega_{n,4} \approx 4.3$ kHz, we observe a weak beat tone from a detuned resonance with an effective Rabi frequency $\Omega_{n,4}^{eff} \approx 10.3$ kHz. From this we conclude a respective detuning of $\Delta \approx 9.4$ kHz which could be caused from the next subset of weaker coupled nuclear spins with a dominant coupling of roughly $A_{\parallel,\uparrow}^5 \approx 2((18.9 + 18.6)/2 - 9.4) \approx 18.6$ kHz which are partially polarized during the SWAP sequence.

This data and the discussion was not yet presented in the manuscript but is now presented in Supplementary Note 10. In addition, we added the following sentence in the “Radio frequency nuclear spin control” section:

“Using the same method we extract $A_{\parallel}^4 = 34$ kHz, inline with previous low-power Rabi measurements on the electron spin.”

3. The demonstrated register relies on three strongly coupled ^{13}C nuclear spins in a specific nanodiamond, which is impressive but highly sample-dependent. The authors should discuss possible routes to scalability. It would also be useful if they could comment on how their strongly coupled register compares conceptually with approaches based on more weakly coupled or isotopically engineered nuclear spins that have been explored in the field.

We agree with the reviewer that a discussion on future scalability is necessary. To this end, we now added a Supplementary Note 3, where we performed Monte Carlo simulations to determine the probability of finding a usable nuclear spin register with 1, 2 or 3 strongly coupled spins. The result is 52.10%, 1.44%, 0.05%, respectively, given our definition of a usable register.

The simulation also show that reducing the inhomogeneous linewidth can drastically improve the register size, from 10^{-2} % to 10^0 % when halving the electron spin's linewidth for a 3-qubit register.

We think that for example cooling the nuclear spin bath using global polarization techniques like NOVEL [SM15] or PulsePol [SM16], SWAP or projective measurements can achieve this goal such that a higher number of strongly coupled nuclear spins become available and addressable with our presented approach.

As the Reviewer already mentioned, using only one strongly coupled nuclear spin is orders of magnitude more likely (>50% probability) and by overcoming the electron spins limited coherence, we can address these weakly coupled nuclear spins with techniques like SEDOR, which was proven successful very recently [7] as well as in our work. With this approach a fault-tolerant register size of three or even more nuclear spins becomes feasible. However, we would like to note that two-qubit gate times are limited by the coupling strength between spins and therefore stronger coupled spins are more favorable from a practical perspective.

As a last remark, we think that since the SiV is hosted in a nanodiamond, we are able to post-select suitable SiVs independently of the coupling to an optical resonator which is crucial for a future quantum

network node. Hence, we think that this system provides an interesting route to a hybrid fault-tolerant network node.

Regarding this discussion, we added a reference to the Supplemental Note 3 in section “Spin characterization of a quasi-free electron”:

“In order to estimate the probability of finding a resolvable nuclear-spin configuration with variable size, we used Monte-Carlo simulations in Suppl. Note 3.”

Reviewer #2 (Remarks to the Author):

Thanks to the high strain property of SiV sample, and the low noise experimental setup, the authors are able to directly drive the nuclear registers using radio wave, instead of via dynamical decoupling which is limited by electron spin coherence time. This pathway enables the author to characterise the nuclear spins more thoroughly, and demonstrate Bell state entanglement.

While my overall impression of the manuscript is positive, I do have a few minor suggestions for the authors to improve it.

Before answering point by point, we would like to express our appreciation of the Reviewers overall suggestions, since we think this helped the manuscript's quality.

I appreciate the authors taking the time to address all of my previous concerns point by point, and make major changes to the manuscript and the supplementary materials to accommodate my comments and suggestion.

1. When reading through the abstract and Introduction, I was confusing by how many nuclear spins are there, as it ranges between two and four, eg. "entanglement of two ^{13}C nuclear spins", "directly detect and control up to four coupled nuclear spins", " 2π - rotation of the electron spin conditioned on three nuclear spins". Fig 1 is not referenced by the main text, but it is actually useful to the user when reading through the Introduction, so I recommend the author to reference the figure with qubit registered n1-4 labelled.

We agree with the Reviewer and modified the introduction, inline with suggestions from Reviewer 1 and 3, to make it clearer that we are controlling a three-qubit nuclear spin register with an auxiliary fourth nuclear spin, which is not used within the register. Additionally, we are directly referencing Fig.1 in that context.

Thank you for modifying the introduction to bring focus to three-qubits nuclear register, while still mentioning the other experiments with less confusion.

2. For each of the [SI], can the author to a specific section or figure within the document?

We are now referencing the [SI] more precisely with the specific section within the main manuscript.

Thank you for the additional reference.

3. In page 3, '..entanglement of that nuclear spin with the electron spin leads to a loss of coherence...', I recommend modifying it to clarify which spin is losing coherence, eg. '... loss of electron coherence...'

We thank the reviewer for the suggestion and modified the respective sentence accordingly.

“Under the right conditions, i.e. the right inter-pulse spacing τ and number of π -pulses N in the pulse sequence depicted in Fig.3a, entanglement of that nuclear spin with the electron spin leads to a loss of electron spin coherence [17,23].”

Thank you for the modification

4. Please indicate which nuclear spin Fig 3(c) refers to.

Fig.3c is addressing all nuclear spins, especially oscillations of the two strongest nuclear spins are visible, highlighted now with white dashed lines. We added the nuclear spin illustrations to Fig.3b&e to make a proper visual connection between the plots.

5. In page 3 of main text, the author showed a set of 4 numbers $A^i_{\perp}/2\pi$, implying the dynamically decoupling experiment was performed on nuclear spins n_1 - n_4 . However, in each of the sub-figure in Fig. 3, the authors chose various number of qubits data to present, with no consistency. Eg. Fig 3(c) &(g) only shows n_1 , fig 3 (a),(b)&(e) shows both n_1 & n_2 , fig 3(f) shows n_1 - n_3 . This makes the reader hard to keep track of which qubit the experiment is performed on. I recommend the authors to at least push the n_3 data in fig 3(f) to supplementary, and add the equivalent fig 3 of qubits n_3 and n_4 , if available, in the supplementary section.

We thank the Reviewer and in addition to the modifications/clarifications of the Reviewer’s 4th comment we now pushed Fig3.f (old) to the Supplementary Note 7.

We extracted the set of four coupling components from the 2D-dataset presented in Fig.3c, where all coupled spins contribute. However, we could not individually resolve coherent oscillations of n_3 and n_4 as we were able to do for n_1 and n_2 in Fig.3b.

Point 4 & 5: The message on the figure is much clearer now, with a strong focus on n_1 & n_2 . I understand by nature n_3 & n_4 are less coupled to the rest of the system, which restricts the extension of experiments the authors can perform on them.

6. Page 4, off-resonance DD:

- By nature, the DD measurement performed here in Fig 3d is very similar to the the CPMG experiment above, the author should comment on the two numbers
- T_2 of 2.31ms is on the same order of magnitude as the oscillation frequency in right of fig 3 b), which is concerning. Can the author points out this (and any other limitation) arise from the T_2 time?

In the updated manuscript the labeling of Fig.3d is now Fig.3f.

a: We thank the reviewer for bringing up a potentially helpful comparison between the two measurements. We would like to mention that the CPMG measurements in Fig.2b is physically different from off resonant DD measurements in the following sense. The CPMG measurements on the one hand sweep the inter-pulse spacing, thus probing the surrounding noise spectrum at different frequencies (for a superset of order N). Given the scaling χ we can extrapolate $T_{2,e}^{CPMG,274}$ for 274 decoupling pulses to $T_{2,e}^{CPMG,274} = T_{2,e}^{CPMG,32} \cdot (274/32)^\chi \approx 3.9ms$, i.e. higher than what we have measured in Fig.3d (now f).

The order sweep in Fig.3d (now f) on the other hand, probes the noise's width at a specific frequency component by keeping the inter-pulse spacing fixed and basically narrowing the filter-functions bandwidth. The discrepancy between CPMG and XY might thus be attributed to specific noise characteristics at $\tau = 8.43\mu\text{s}$ or remaining pulse errors which are not compensated at very large N. The latter might not have been visible in CPMG measurements. Both causes can be investigated in future works but are not the focusing here.

For clarification and deeper insights into the presented measurements, we now added an additional sentence with the comparison presented here since. Also, we moved the sentence to the end of the DD section, for a better flow of information. Inline with comment 8 of Reviewer #1 we added:

"In order to probe the electron spin's coherence under XY-N DD we choose $\tau = 8.43\ \mu\text{s}$, off-resonant with a multiple of the nuclear spins precession period, sweep N and observe an exponential decay in coherence within $T_{2,e}^{XY} = 2.31(14)\text{ms}$, shown in Fig.3f. The measured $T_{2,e}^{XY}$ poses an ultimate limit to our sensitivity of detecting nuclear spins with DD and limits indirect nuclear spin control gates, as can be seen by the exponential decaying oscillations in Fig.3b. It is also worth noting that $T_{2,e}^{XY}$ differs from an extrapolation of the CPMG measurements, which yields $T_{2,e}^{CPMG,274} = T_{2,e}^{CPMG,32} \cdot (274/32)^{\chi} \approx 3.9\text{ms}$. The discrepancy is attributed to the specific noise characteristics at $\tau = 8.43\ \mu\text{s}$ or pulse errors at large $N \gg 32$."

I appreciate the authors detail explanation of the distinguishment of CPMG and XY, and amend the manuscript text to clarify them. It is helpful for me to think in the frequency domain. With a selection of τ and N, the 'true' limit of decoherence of the electron is explored.

7. In page 4, tau_init was used without being defined. Is it equivalent to tau in Fig 3(a)?

We appreciate the Reviewer's comment on a missing proper definition of τ_{init} , which was indirectly defined in the caption of Fig.3f (old). To improve clarity and in conjunction with the Reviewers suggestion 5, we now modified the manuscript and referenced a section to the Supplementary Figure 5:

"[...] requiring electron spin conditioned $\pi/2$ rotations of the target nuclear spin with a slightly off-resonant τ_{init} , see the sequence depicted in Fig.3d. We implement the corresponding sequence with $N = 24$ and sweep τ_{init} around the resonance, where individual initialization is achieved at $\tau_{init,1} = 8.3915\ \mu\text{s}$, see Suppl. Fig. 5."

After reading the updated main text, I was still confused by the relationship or difference between τ_{init} and τ . However, I'm glad that that the authors refers the reader to Suppl. Fig. 5. Alongside with fig 3a and d, It is now clear to me that τ_{init} is choosing a fix value of τ in the circuit of Fig 3d, which refers to Fig 3a.

8. The manuscript indicates individual initialization is achieved at $t_{init,1} = 8.3915\ \mu\text{s}$, as indicated in Fig 3f. Is the criteria maximal $\langle \rho_z \rangle$? If yes, then why is 8.3915us chosen over 8.3955us?

We thank the Reviewer for this observation and would like to address it with the following argument. Since the nuclear spin's initialization and readout fidelity with this sequence is limited by the additional nuclear spins we are not able to differentiate whether any nuclear spin is better or worse initialized than

the other. In order to independently measure the initialization fidelities, depicted in Supplementary Fig. 5, one would have to use a different measurement technique, see for example [18], Fig.3c.

We chose the resonance, because our numerical model predicts for this exact resonance polarization of only one nuclear spin, facilitating the interpretation of the measured Ramsey signal. We added in Supplementary Note 7 a discussion about the choice of τ_{init} .

Thank you for highlighting the limitation of multiple nuclear spins in the experimental setup. The single sinusoidal functional in fig 3e also support n1 being the dominate spin at the chosen τ_{init} ,

9. In Fig 5b, the boundaries of the shade regions are pixelated. Is it an artefact or does it indicates the frequency resolution of the sinc function?

The pixelated-looking boundary is indeed due to the limited frequency resolution set by the truncated sinc function in the time-domain.

I see. Thank you for clarifying.

10. In page 6, why a pulse of 150kHz is chosen? given that the smallest hyperfine energy splitting is ~ 141 kHz (n_3), a pulse of 150kHz doesn't seem to have sufficient spectral resolution to distinguish spin up and down of n_3 .

We thank the Reviewer for bringing up this concern. In an ideal scenario, i.e. infinitely temporal extension of the pulse, the resonant 150 kHz sinc-pulse's spectral window would not cause spectral leakage into the next pair of resonances, making it a conditional pulse. This is due to the fact the resonances of n_3 are split by 141 kHz such that the pulse's bandwidth only accommodates roughly half the splitting, namely $150 \text{ kHz}/2 = 75 \text{ kHz} < 141 \text{ kHz}$. However, due to our truncation and the aforementioned frequency resolution there is some expected small spectral leakage which we simulated in the Supplementary Note 9, when considering the amplitude sweeps to implement CNOTs.

Reducing the bandwidth would've helped considering the mentioned argument but would then have caused issues with the next weaker coupled nuclear spins, which have to be addressed unconditionally. For this reason, we chose 150 kHz. Further experimental optimization of the gate fidelity regarding truncation and bandwidth or a different pulse-shape altogether could be an object of future experiments.

Thank you for the response, and pointed out that only half of the pulse bandwidth should be consider when comparing with the 141kHz splitting. And indeed, given that n_4 has hyperfine splitting of 34kHz plus linewidth, smaller pulse bandwidth will risk conditional rotation of n_4 , as well as concern on spectral resolution

11. In page 7, has the authors attempt prepare other 3 Bell states? If so, what is the fidelity? Depends on the locality of the noise source, parallel Bell states might yield higher state fidelity.

After a minor revision stated above, I would like to recommend an acceptance of this manuscript to Nature communications.

We thank the Reviewer for bringing up this interesting insight. We indeed measured the other three Bell states. The one we are presenting currently in the main manuscript turned out to have the highest fidelity. For the data of the other Bell states with discussion, the Reviewer is referred to the new

Supplemental Note 12.

Thank you for adding the other 3 Bells state results to the Supplementary notes to form a more completed picture in terms of demonstrating entangled states.